# Nanopore analysis of salvianolic acids in herbal medicines

Pingping Fan[1,2,5], Shanyu Zhang [1,2,5], Yuqin Wang [1,2,3,4], Tian Li[1,2], Hanhan Zhang[1,2], Panke Zhang [1] & Shuo Huang [1,2] ✉

Natural herbs, which contain pharmacologically active compounds, have been used historically as medicines. Conventionally, the analysis of chemical components in herbal medicines requires time-consuming sample separation and state-of-the-art analytical instruments. Nanopore, a versatile single molecule sensor, might be suitable to identify bioactive compounds in natural herbs. Here, a phenylboronic acid appended *Mycobacterium smegmatis* porin A (MspA) nanopore is used as a sensor for herbal medicines. A variety of bioactive compounds based on salvianolic acids, including caffeic acid, protocatechuic acid, protocatechualdehyde, salvianic acid A, rosmarinic acid, lithospermic acid, salvianolic acid A and salvianolic acid B are identified. Using a custom machine learning algorithm, analyte identification is performed with an accuracy of 99.0%. This sensing principle is further used with natural herbs such as *Salvia miltiorrhiza*, *Rosemary* and *Prunella vulgaris*. No complex sample separation or purification is required and the sensing device is highly portable.

Herbal medicines, also known as phytomedicines, are mixtures of plant metabolites that contain pharmacologically active compounds with some therapeutic properties[1]. Herbal medicines have been widely used historically for the treatment of numerous diseases, including but not limited to plague[2–4], cardiac cerebral disease[5,6] and pain relief[7,8]. The therapeutic effects of herbal medicines are mainly attributable to their bioactive compounds, which are natural products, such as the anti-malarial artemisinin from *Artemesia annua*[9,10], the anti-inflammatory aspirin from *Salix*[7] and the analgesic morphine from *Papaver somniferum*[8]. Accurate identification and quantification of the bioactive compounds in herbal medicines is essential for the discovery, production and quality control of herbal medicines[11]. However, due to their complexity, quantitative analysis of natural herbs is non-trivial.

Conventionally, Fourier transform infrared spectroscopy (FTIR)[12,13], high-performance liquid chromatography (HPLC)[14] and liquid chromatography-mass spectrometry (LC-MS)[15] have been applied to the quantitative analysis of herbal medicines but these spectroscopic and chromatographic techniques usually require complicated and time-consuming sample pretreatment. With HPLC, simultaneous detection of multiple kinds of components in herbal medicines under the same measurement condition is difficult[16]. The overlap in FTIR spectra[13] and the complexity of LC-MS data[17] also pose challenges for data interpretation. Besides, the required state-of-the-art instruments are generally bulky and expensive and are not suitable for direct and rapid analysis of natural herb samples in a field environment.

Biological nanopore, originally developed for nucleic acid sequencing, is a versatile single molecule sensor of nucleic acids[18,19], proteins[20–22] and small molecules[23,24]. When appropriately modified with a reactive adapter, a biological nanopore becomes a nanoreactor, which is responsive to chemically compatible small molecules[25–27]. Under sensing in a nanopore, a target analyte specifically binds to and dissociates from the reactive adapter, producing highly characteristic nanopore events resulting from dynamic single molecule reactions

[1]State Key Laboratory of Analytical Chemistry for Life Sciences, School of Chemistry and Chemical Engineering, Nanjing University, 210023 Nanjing, China. [2]Chemistry and Biomedicine Innovation Center (ChemBIC), Nanjing University, 210023 Nanjing, China. [3]State Key Laboratory of Pollution Control and Resource Reuse, School of the Environment, Nanjing University, 210023 Nanjing, China. [4]Institute for the Environment and Health, Nanjing University Suzhou Campus, 215163 Suzhou, China. [5]These authors contributed equally: Pingping Fan, Shanyu Zhang. ✉e-mail: shuo.huang@nju.edu.cn

occurring in the pore lumen. This mode of sensing is advantageous for recognition of target analytes directly from a complex mixture of compounds, since the interfering molecules in the environment either generate events that are readily distinguishable from the target analytes or fail to generate any events. Natural herbs, which contain both the desired bioactive components and interfering background analytes, are suitable for nanopore sensing. Compared with conventional herb analysis methods[12,16,17], nanopore sensing has the advantages of a single molecule resolution, high accuracy, a facile sample separation and a high portability. To the best of our knowledge however, nanopore analysis of herbal medicines has never been previously demonstrated.

As has been recently reported, a *Mycobacterium smegmatis* porin A (MspA) nanopore modified with phenylboronic acid (PBA), also referred to as MspA-90PBA, demonstrates an exceptional resolution in the identification of *cis*-diols such as monosaccharides[28], alditols[29] and ribonucleotides[30,31]. The PBA adapter also serves to reversibly capture and release the analyte during recording, so that an extended event dwell time is reported. One drawback of this technique is that the PBA adapter is in principle not suitable for *trans*-diols due to the unmatched spatial conformation (Supplementary Fig. 1). Natural herbal medicines contain various *cis*-diol compounds including but not limited to phenolic acids[32–34], saccharides[35,36] and anthocyanin[37,38]. Salvianolic acids are members of a family of the most abundant water-soluble phenolic acids in *Salvia miltiorrhiza* which have been widely discovered in other herbs such as *Rosemary*[39], *Prunella vulgaris*[38] and *Mint*[40]. Naturally occurring salvianolic acids have been widely applied in the treatment of cardiovascular[6] and cerebrovascular[41] disease. According to the literatures[42,43], salvianolic acids, with the exception of

protocatechuic acid (PCA) and protocatechualdehyde (PA), are composed of salvianic acid A (SAA) or caffeic acid (CA) which acts as basic building blocks. This suggests that almost all salvianolic acids contain 1, 2-diol groups and can be detected by MspA-90PBA.

In this work, a single phenylboronic acid appended MspA (MspA-90PBA) nanopore (Fig. 1a) is used for the identification and quantification of salvianolic acids in herbal medicines, including injections and natural herbs. Assisted by a custom machine learning algorithm, the high resolution of MspA allows full discrimination between different types of salvianolic acids in natural samples. Moreover, this sensor can also be further integrated into a portable device to assist natural product investigations during fieldwork or for extreme situations. By being equipped with other chemical modifications, this nanopore sensor may also be suitable for a wider variety of herb samples.

## Results

### Identification of different salvianolic acids using MspA-90PBA

Unless stated otherwise, all nanopore measurements were performed using MspA-90PBA (Fig. 1a) in a buffer of 1.5 M KCl, 100 mM MOPS, pH 7.0. A + 100 mV bias was continually applied (for details, see Methods). In principle, all chemical components containing a *cis*-diol structure should react with the phenylboronic acid (PBA) adapter of MspA-90PBA to produce nanopore events[28,44]. If a molecular analyte contains multiple *cis*-diol structures, it would be expected to report multiple types of events resulting from different modes of binding. Compounds which fail to react with MspA-90PBA are not reported during nanopore sensing and this chemical selectivity enables recognition of target analytes directly from a mixture of compounds without any need for complex sample pretreatment.

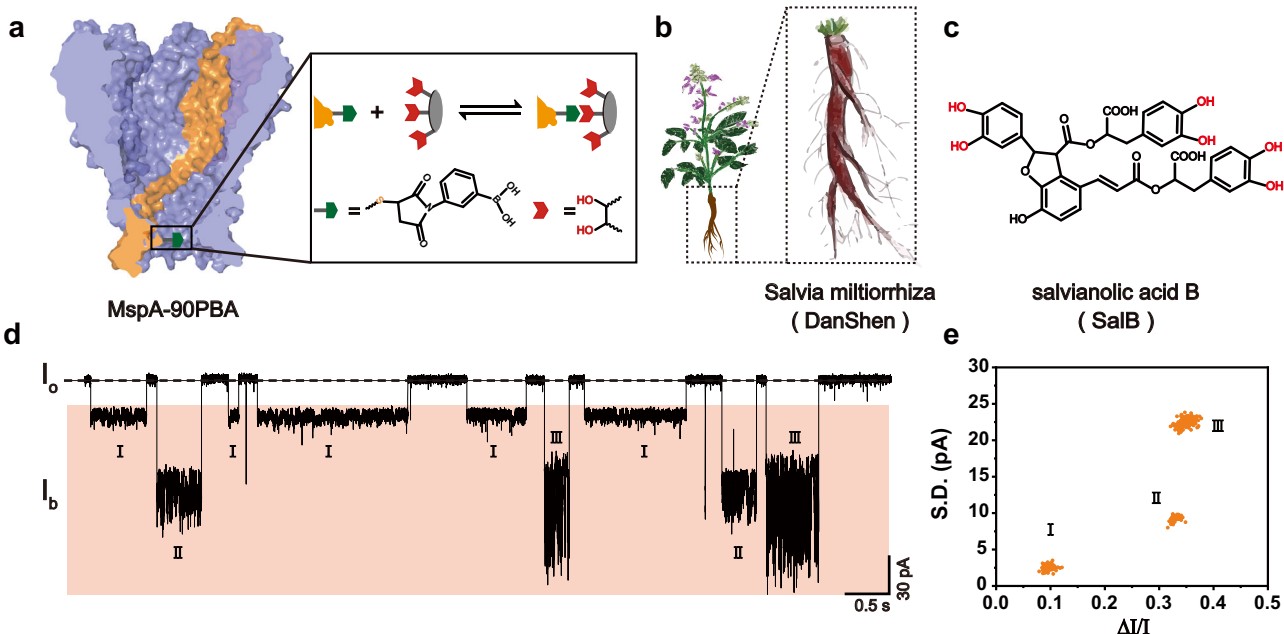

**Fig. 1 | Identification of salvianolic acid B using MspA-90PBA. a** The structure of MspA-90PBA. MspA-90PBA is a hetero-octameric MspA modified with a single phenylboronic acid (PBA) adapter at its pore constriction (**Methods**). The mechanism of salvianolic acid sensing is described on the right. Briefly, the PBA at the pore constriction can react reversibly with a *cis*-diol group of the analyte to produce a nanopore event. **b** A cartoon of the herb *Salvia miltiorrhiza Bunge*. *Salvia miltiorrhiza* (*Danshen*, dotted box), which is the root of *Salvia miltiorrhiza Bunge*, and contains rich levels of salvianolic acids. **c** The chemical structure of salvianolic acid B (SalB). SalB is a type of salvianolic acids. The 1, 2-diol groups on SalB are marked in red. SalB, which is the most abundant salvianolic acid in *Salvia miltiorrhiza*, is widely applied in the treatment of cardiovascular and cerebrovascular

diseases. **d** A representative trace containing nanopore events of SalB. The measurement was carried out using MspA-90PBA in a buffer of 1.5 M KCl, 100 mM MOPS, pH 7.0. A + 100 mV voltage was continually applied. SalB was added to *cis* with a final concentration of 0.1 mM. Three types of events were observed from the trace. For the ease of demonstration, each event was respectively marked with different roman numerals to show their identities. **e** The scatter plot of $\Delta I/I_o$ versus *S.D.* for data acquired as described in (**d**). To remove background noises, the data in the scatter plot was treated by cluster analysis using DBSCAN (Supplementary Fig. 4). 672 events were demonstrated in the scatter plot. Source data are provided as a Source Data file.

In traditional Chinese medicine, *Salvia miltiorrhiza* (*Danshen*) has been widely used to treat cerebrovascular diseases[41] (Fig. 1b). In later investigations, salvianolic acid B, ((2*R*)-2-[(*E*)-3-[(2*R*,3*R*)-3-[(1*R*)-1-carboxy-2-(3,4-dihydroxyphenyl)ethoxy]carbonyl-2-(3,4-dihydroxyphenyl)-7-hydroxy-2,3-dihydro-1-benzofuran-4-yl]prop-2-enoyl]oxy-3-(3,4-dihydroxyphenyl)propanoic acid), or SalB, the most abundant salvianolic acid in *Salvia miltiorrhiza*, was identified as the effective compound[45] (Fig. 1c). The chemical structure of SalB includes three separate 1, 2-diol structures, enabling its interactions with MspA-90PBA. In view of the high resolution of MspA, three discriminable event types which respectively originate from different binding modes of SalB with PBA, may also potentially be observed.

To support this, a nanopore measurement was arranged and SalB was added to the *cis* chamber at a final concentration of 0.1 mM (Methods). Corresponding nanopore events were observed immediately (Fig. 1d). To describe these nanopore events quantitatively, the relative blockage depth $\Delta I/I_o$ $((I_o-I_b)/I_o)$, standard deviation *S.D.*, dwell time $t_{off}$ and the inter-event interval $t_{on}$, are defined as shown in Supplementary Fig. 2. The reciprocal of the inter-event interval $(1/\tau_{on},$ N = 3) is linearly correlated with the SalB concentration, which is consistent with a bimolecular model[24,46] (Supplementary Fig. 3 and Supplementary Table 1). The event dwell time ($\tau_{off}$, $N = 3$) however, is independent of the SalB concentrations, which is consistent with a unimolecular dissociation mechanism (Supplementary Fig. 3 and Supplementary Table 1). The kinetics further confirmed that the observed events were from SalB. According to event features such as $\Delta I/I_o$ and *S.D.*, three types of events can be clearly observed from the trace (Fig. 1d), and this is more clearly demonstrated in the scatter plot of $\Delta I/I_o$ versus *S.D.* (Fig. 1e and Supplementary Fig. 4). To avoid interference from noise events, density-based spatial clustering of applications with noise (DBSCAN), a cluster analysis algorithm, was applied to remove events which fail to form a clear cluster (Supplementary Fig. 4). The ratios of events removed by DBSCAN are summarized in Supplementary Table 2. The count of event clusters is consistent with the number of 1, 2-diol groups of SalB (Fig. 1c), suggesting that each type of event results from a specific binding mode between SalB and PBA.

This sensing principle was also applied to other salvianolic acids including caffeic acid (CA), protocatechuic acid (PCA), protocatechualdehyde (PA), salvianic acid A (SAA), rosmarinic acid (RA), lithospermic acid (LSA), salvianolic acid A (SalA) and salvianolic acid B (SalB)[47,48] (Fig. 2a–h). All aforementioned salvianolic acids report unique nanopore events. The count of event types is generally consistent with the number of 1, 2-diol structures of the salvianolic acids being tested, further confirming that the different event types are a result of different binding modes between components of salvianolic acid and PBA. Salvianolic acids containing only a single 1, 2-diol structure, such as CA, PCA and PA, only report a single type of event for each analyte. The scatter plots generated by each type of salvianolic acid are also shown in Supplementary Figs. 4–11. The ratios of interference events removed by DBSCAN are summarized in Supplementary Table 2. The interference events can be from impurities in the analytes derived from plant extraction[49] or chemical degradation of salvianolic acids[50–53]. Spontaneous pore gating also contributes to the generation of interference events as well. Representative traces acquired with different salvianolic acids and the corresponding scatter plots are also demonstrated in Supplementary Figs. 12, 13 to show the data quality. When simultaneously compared in the same scatter plot of $\Delta I/I_o$ versus *S.D.*, nanopore events acquired from different salvianolic acids are clearly distinguishable (Fig. 2i, Supplementary Table 3). To this end, MspA-90PBA has shown direct recognition of up to eight salvianolic acids whose event features are well discriminated by simultaneously considering the event features of $\Delta I/I_o$ versus *S.D.*. When captured by the PBA adapter and chemically confined in the pore lumen, the analyte may further interact with the amino acid

residues of the pore to produce characteristic noises on top of the blockage levels, which is useful for event identification. Salvianolic acids containing multiple 1, 2-diol structures also report multiple event types, and these event types are also distinguishable in the corresponding scatter plot of $\Delta I/I_o$ versus *S.D.*, acknowledging the high resolution of this engineered MspA sensor.

## Machine learning assisted identification of salvianolic acids
In the field of nanopore research, machine learning has been widely applied to assist data analysis of nucleic acid sequencing[54,55] and single molecule sensing[56,57]. In this work, the data acquired with complex samples leads to the difficulty of event identification by human eyes. Besides, when a large amount of data is involved, event identification automation also becomes urgent. To also quantitatively assist event identification and sensing performance evaluation, a Python-based machine learning algorithm was developed. At the outset, 500 events acquired with each type of salvianolic acids were selected for feature extraction. Different event types generated by the same type of salvianolic acid were not differently labeled during the training process. The event features are also independent of the concentration of the analyte used to produce the event.

A total of 4000 events from all eight types of salvianolic acids were collected and two event features, including $\Delta I/I_o$ and *S.D.*, were used to build a feature matrix (Fig. 3a and Supplementary Fig. 14). The event label was assigned with the salvianolic acid that generated the event and in this way, a dataset was formed. This dataset was then randomly split into a training set (80% of the dataset) for model training and validation and a testing set (20%) for model testing. Six commonly used models[28,30], including K-NearestNeighbor (KNN), Extreme Gradient Boosting (Xgboost), Classification and Regression Tree (CART), Support Vector Machine (SVM), Gradient Boost Decision Tree (GBDT) and Random Forest (RF) were applied for model training and default model parameters were used. The validation accuracy derived from the results of a 10-fold cross validation was used to identify the best performing model. KNN, which produces a 99.0% validation accuracy, is the best performing model (Fig. 3b). All models report a high classification accuracy, suggesting that the data acquired with different salvianolic acids was easily discriminable. All the above trainings were carried out with the default hyperparameter settings.

The confusion matrix generated by prediction of the testing set using the previously trained KNN model is also shown in Fig. 3c (Methods). To evaluate the efficiency of a model, the learning curve was also plotted and no overfitting was seen. According to this learning curve, the validation score reached 0.984 when 291 training samples were included (Supplementary Fig. 15).

The trained KNN model was then employed to predict unlabeled events during simultaneous sensing of all eight salvianolic acids. The measurement was carried out as described in Fig. 3d. In the representative trace, characteristic nanopore events of all eight types of salvianolic acids were clearly seen based on their event features, consistent with the results produced when different salvianolic acids were tested separately (Supplementary Figs. 12, 13). All acquired nanopore events described in Fig. 3d were collected to perform feature extraction (Supplementary Fig. 14). To reduce interference of non-clustered background noises, the events were further treated by DBSCAN (Supplementary Fig. 16). Subsequently, the events were directly transmitted to the previously trained KNN model for event identification. All identified events were labeled on the trace (Fig. 3d, Supplementary Movie 1) or color labeled in the corresponding scatter plot (Fig. 3e, Supplementary Fig. 17). Although only two event features including $\Delta I/I_o$ and *S.D.* were employed for machine learning, the produced prediction accuracy is sufficiently good based on results shown in model validation (Fig. 3b), confusion matrix (Fig. 3c), learning curve production (Supplementary Fig. 15) and simultaneous analyte sensing and prediction (Fig. 3d, e). These results suggest that the event

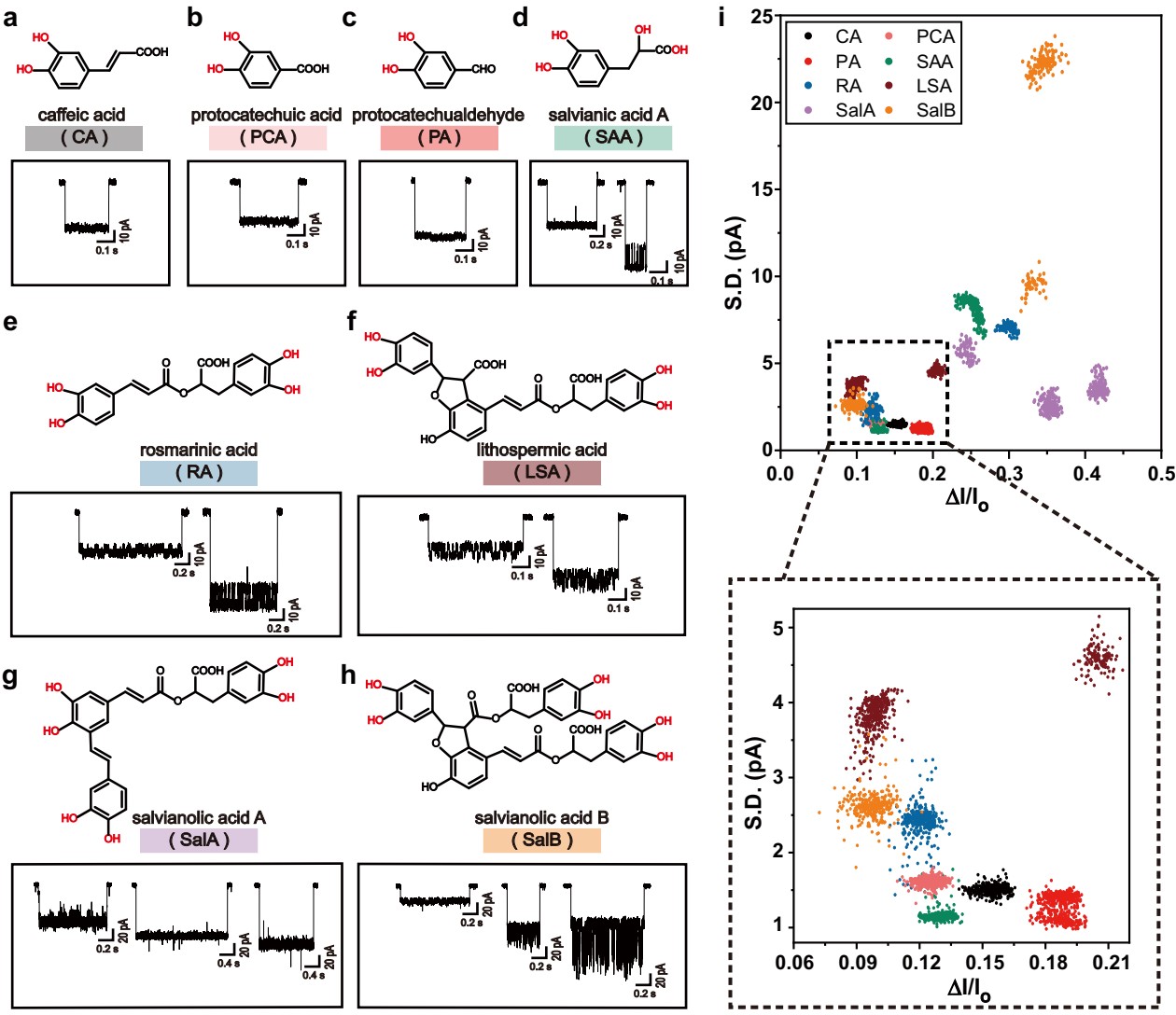

**Fig. 2 | Discrimination of eight salvianolic acids using MspA-90PBA. a–h** The chemical structures of eight types of salvianolic acids and their corresponding nanopore events. The salvianolic acids include caffeic acid (CA), protocatechuic acid (PCA), protocatechualdehyde (PA), salvianic acid A (SAA), rosmarinic acid (RA), lithospermic acid (LSA), salvianolic acid A (SalA) and salvianolic acid B (SalB). The 1, 2-diol groups of each compound are marked in red. The abbreviations of each analytes are also marked with color bands, including black (CA), pink (PCA), red (PA), green (SAA), blue (RA), wine-red (LSA), lavender (SalA) and orange (SalB). All measurements were carried out using MspA-90PBA in a buffer of 1.5 M KCl, 100 mM MOPS, pH 7.0 (Methods). CA (1 mM), PCA (2 mM), PA (0.5 mM), SAA (0.5 mM), RA (0.3 mM), LSA (0.2 mM), SalA (0.03 mM) and SalB (0.1 mM) were separately added to *cis*. A + 100 mV bias was continually applied. CA (**a**), PCA (**b**) and PA (**c**) contain a single 1, 2-diol group and only one type of event was reported for each type of analyte. SAA (**d**), RA (**e**) and LSA (**f**), which contain two 1, 2-diol groups, report two types of events. SalA (**g**) and SalB (**h**), which contain three 1, 2-diol groups, report three types of events. (**i**) **Top**: The scatter plot of $\Delta I/I_o$ versus *S.D.* of events acquired from all eight types of salvianolic acids. 500 events acquired with each type of analyte were included in the scatter plot (n = 4000). To remove background nomises, all events were treated by cluster analysis using DBSCAN, as described in Supplementary Figs. 4–11. **Bottom:** the zoomed-in view of the area marked with a dashed box in the top. Source data are provided as a Source Data file.

features of all eight salvianolic acids can be significantly discriminated from one another and the measurement consistency between pores is satisfactory. Considering the influence of different features on the prediction results, the importance of $\Delta I/I_o$ and *S.D.* are evaluated for eight analyte discrimination. The superimposed histograms of SalB, SalA and RA were shown in Supplementary Fig. 18 and demonstrated the fact that some salvianolic acids are indistinguishable when either $\Delta I/I_o$ or *S.D.* was employed. Furthermore, the mutual information between these two features were also calculated (Supplementary Fig. 18). Here, the mutual information value measures the correlation between the features and event labels, where a higher value indicates a closer correlation and the more important this feature is. The values of the two features are similar, indicating the equal importance of $\Delta I/I_o$ and *S.D.* in event identification. Definitely, more event features could

be further included in the machine learning model building when additional types of events need to be simultaneously distinguished.

## Rapid identification of salvianolic acids from salvianolate injection

The above demonstrated sensing capacity and the custom data analysis algorithm can be used for rapid identification of salvianolic acids components directly from an actual biological sample. Salvianolate injection is a natural herbal medicine derived from extracts of *Salvia miltiorrhiza* for the clinical treatment of coronary heart disease[58,59]. Although the magnesium salt of SalB is the main bioactive component in the salvianolate injection[45], the production of salvianolate injection[45,60] which includes extraction of *Salvia miltiorrhiza* with ethanol or hot water, macroporous resin adsorption, ethanol gradient

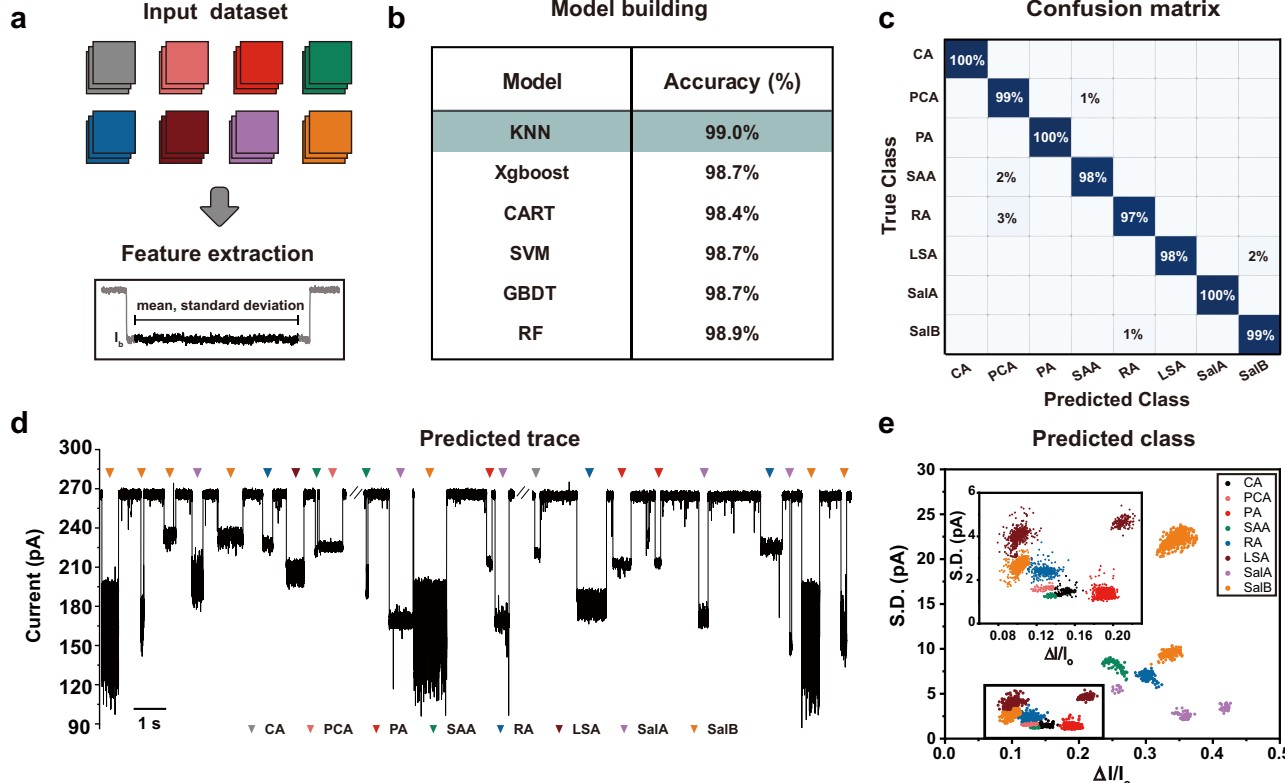

**Fig. 3 | The machine learning workflow. a** The training dataset. 500 events acquired with each type of analytes, including CA, PCA, PA, SAA, RA, LSA, SalA and SalB, were collected to form the training dataset (top). Two event features, including the relative blockage depth ($\Delta I/I_o$) and the standard deviation (*S.D.*), were extracted from each event to form a feature matrix (bottom). **b** Training accuracies. Six commonly used models including K-Nearest Neighbor (KNN), Extreme Gradient Boosting (Xgboost), Classification and Regression Tree (CART), Support Vector Machine (SVM), Gradient Boost Decision Tree (GBDT) and Random Forest (RF) were evaluated. KNN, which reports the highest validation accuracy, was selected for all subsequent prediction tasks. **c** The confusion matrix result of salvianolic acids classification performed by the trained KNN model. **d** A representative trace acquired by simultaneous sensing of all eight salvianolic acids. The measurement was carried out using MspA-90PBA in a buffer of 1.5 M KCl, 100 mM MOPS, pH 7.0 (Methods). All analytes were added to *cis* to reach the desired final concentrations and a +100 mV bias was continually applied. Specifically, the final concentrations of CA and SAA were 40 μM, that of PCA was 100 μM and that of PA, RA, LSA, SalA and SalB were 20 μM. All events were automatically predicted by machine learning and labeled with corresponding labels. **e** The scatter plot of $\Delta I/I_o$ versus *S.D.* of results acquired by simultaneous sensing of all eight salvianolic acids using the same nanopore (n = 4268). Each event was identified by the previously trained KNN model and is color labeled. Source data are provided as a Source Data file.

elution, concentration and drying, resulting in multiple types of salvianolic acids in the injection sample. This suggests that a nanopore assay may assist in the characterization of the SalB components in the salvianolate injection sample and could provide a standard of medicinal quality control.

Nanopore measurements with salvianolate injection was carried out as shown in Fig. 4a. After setting up the measurement, a 4 μL injection sample was directly added to the *cis* side, without performing any sample pretreatment. Immediately after this, successive resistive pulses were observed in the nanopore trace (Fig. 4b). The event features of all nanopore events were extracted to generate the scatter plot and background noise reduction was performed by the previously described DBSCAN algorithm (Supplementary Fig. 19). The corresponding ratio of interference events removed by DBSCAN is demonstrated in Supplementary Table 2. After noise reduction, a scatter plot of $\Delta I/I_o$ and *S.D.* of results acquired with the salvianolate injection was generated (Fig. 4c). All events were predicted by the previously trained KNN model. According to the prediction results, components such as SalB, RA and LSA were clearly identified from the injection sample, and chemical components other than SalB were detectable from the salvianolate injection sample, consistent with results produced by HPLC measurements in previous reports[61,62]. The same conclusion was drawn from results acquired in three independent trials (Supplementary Figs. 19, 20), confirming the reproducibility of the measurement and the validity of the conclusion. Besides SalB, RA and LSA, traces of CA, SAA and PCA were also detected and identified using the previously trained KNN model (Fig. 4d and Supplementary Fig. 20). This result has not been previously reported by HPLC measurements of salvianolate injection, demonstrating the superior resolution and sensitivity of nanopore in the detection of trace amounts of target analytes.

Subsequently, to verify that the measured value of the content of the magnesium salt of SalB was consistent with the amount indicated in the product manual, the SalB content in the injection was further quantified. The calibration curve of pure SalB with a coefficient factor ($R^2 = 0.978$) is shown in Supplementary Fig. 21. From the KNN model prediction, SalB events in the salvianolate injection were identified automatically (Supplementary Fig. 20). The $1/\tau_{on}$ value of SalB was derived (Methods and Supplementary Table 4), according to which, the SalB content in the injection was calculated according to the calibration curve. Based on results of three independent measurements, the mass of the magnesium salt of SalB in the salvianolate injection was derived to be -35.1 ± 0.2 mg (Supplementary Fig. 21 and Supplementary Table 4), consistent with the reference value in the product manual, which is 40 mg (Supplementary Fig. 21). All the above results confirm that the salvianolic acids components in the salvianolate injection can be well characterized by MspA-90PBA. Though the main component of the salvianolate injection was confirmed to be SalB, the nanopore detected other components such as RA, LSA, CA, SAA and PCA. To

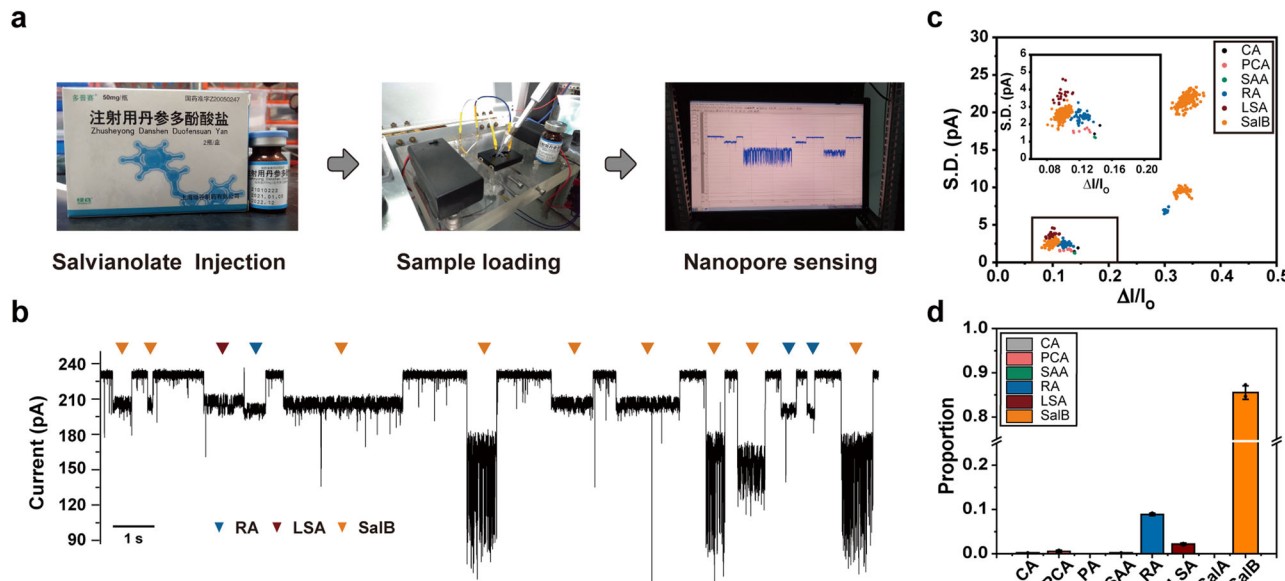

**Fig. 4 | Rapid analysis of salvianolic acids in salvianolate injection. a** The workflow of salvianolate injection analysis. **Left**: The powder of salvianolate injection was dissolved in Milli-Q water to reach a 5 mg/mL concentration. **Center**: 4 μL dissolved salvianolate injection was added to the *cis* chamber of a nanopore device. The measurement was carried out using MspA-90PBA in a buffer of 1.5 M KCl, 100 mM MOPS, pH 7.0 (Methods) and a bias of +100 mV was continually applied. **Right**: Corresponding nanopore events observed immediately. (**b**) A representative trace acquired during salvianolate injection analysis. The events were identified by the trained KNN model and are labeled accordingly. (**c**) The scatter plot of $\Delta I/I_o$ versus $S.D.$ of events acquired with the salvianolate injection.

The events in the scatter plot were taken from a 30 min continually recorded trace and a total of 846 events were collected. The events were labeled according to the prediction results performed by the previously trained KNN model. (**d**) The proportion of salvianolic acid events in the salvianolate injection. Data were presented as mean ± standard deviation values derived from results of three independent measurements (N = 3) (Supplementary Fig. 20). The error bars represent standard deviation values. Clearly, SalB is the main component of the salvianolate injection. However, other salvianolic acid components were also detected by nanopore. Source data are provided as a Source Data file.

verify the effectiveness of the quantitative results, a mixture with predetermined concentrations of eight salvianolic acids was also measured and quantitatively analyzed as described in Supplementary Fig. 22. In principle, the concentrations of all salvianolic acids can be calculated by their corresponding calibration curves as described in Methods. Here, taking SalB as an example (Supplementary Fig. 22), the concentration of SalB in the mixture was derived to be ~ 23.6 μM, highly consistent with actual value, ~ 25 μM, confirming the capacity of nanopore in the quantification of bioactive compounds in herbal medicines. All these results suggest that the demonstrated nanopore assay is potentially suitable for quality control, drug screening or pharmacokinetics analysis of herbal medicines.

**Rapid identification of salvianolic acids in natural herbs**
The sensing configuration described here is also suitable for detection of salvianolic acids directly from natural herbs. *Salvia miltiorrhiza*, the root of *Salvia miltiorrhiza Bunge*, was used historically in China to treat cardiovascular disease[6], and specifically, angina pectoris[63] and myocardial infarction[64]. However, the complex composition of *Salvia miltiorrhiza* may pose challenges for the quality control of the herbal medicine and in addition, its clinical efficacy and safety cannot be guaranteed. The chemical constituent in *Salvia miltiorrhiza* has been studied extensively for over 80 years[65]. According to published reports[34,48,66], salvianolic acids found in extracts of *Salvia miltiorrhiza* include SalB, SalA and RA. In clinical research, salvianolic acids were found to have important pharmacological effects such as scavenging free radicals[67], and affected Na/K-ATPase[68] activity, thereby minimizing cardiovascular and cerebrovascular damage. Although the bioactive constituents of *Salvia miltiorrhiza* have been intensively studied with chromatographic techniques[48,69,70], a rapid single-molecule characterization of *Salvia miltiorrhiza* has never been reported to date.

The general workflow of *Salvia miltiorrhiza* analysis by nanopore is described in Fig. 5a and Supplementary Fig. 23a. Briefly, *Salvia*

*miltiorrhiza* was crushed and soaked in deionized water at 4 °C for 12 h to extract salvianolic acids. The soaking liquid was centrifuged at 4 °C and 1500 *g* for 10 min and then the supernatant was collected and ultrafiltered with a 3 kDa ultrafiltration tube at 4 °C and 1900 *g* for 30 min. Subsequently, 20 μL of the filtrate was added to the *cis* chamber to initiate the measurement (Fig. 5b, c) and data analysis by machine learning was carried out with collected nanopore events. Distinct from reported strategies of herbal medicine analysis[69,71], this strategy does not require any chromatographic methods for separation.

Two features of all nanopore events, $\Delta I/I_o$ and $S.D.$ were extracted from the raw traces to generate the scatter plot (Supplementary Fig. 24). According to published reports[72], there are in *Salvia miltiorrhiza* a variety of salvianolic acids and other substances containing *cis*-diol groups. In order to focus on previously trained salvianolic acids event types, all collected events were further treated by One-Class SVM to remove events that don't belong to any previously trained dataset (Supplementary Fig. 24). The corresponding ratio of removed interference events is summarized in Supplementary Table 2. After the One-Class SVM treatment, all remaining events were predicted by the previously trained KNN model. All identified events were labeled on the trace (Fig. 5c, Supplementary Movie 2) or color marked in the scatter plot (Supplementary Fig. 25a–c), from which SalB, RA, LSA and few of other salvianolic acids could be clearly identified. Three independent trials were carried out (Supplementary Figs. 24, 25), confirming the reproducibility of the measurement and the validity of the conclusion. According to the statistics, the relative content of SalB was identified to be the highest among all eight types of salvianolic acids. This also demonstrates the advantages of *Salvia miltiorrhiza* as the source material for SalB production (Fig. 5d). The $1/\tau_{on}$ values of SalB events were subsequently calculated as described in Methods, according to which, the amount of SalB extracted from *Salvia miltiorrhiza* was derived based on the calibration curve (Supplementary Table 5). Finally, the SalB in *Salvia miltiorrhiza* was calculated to

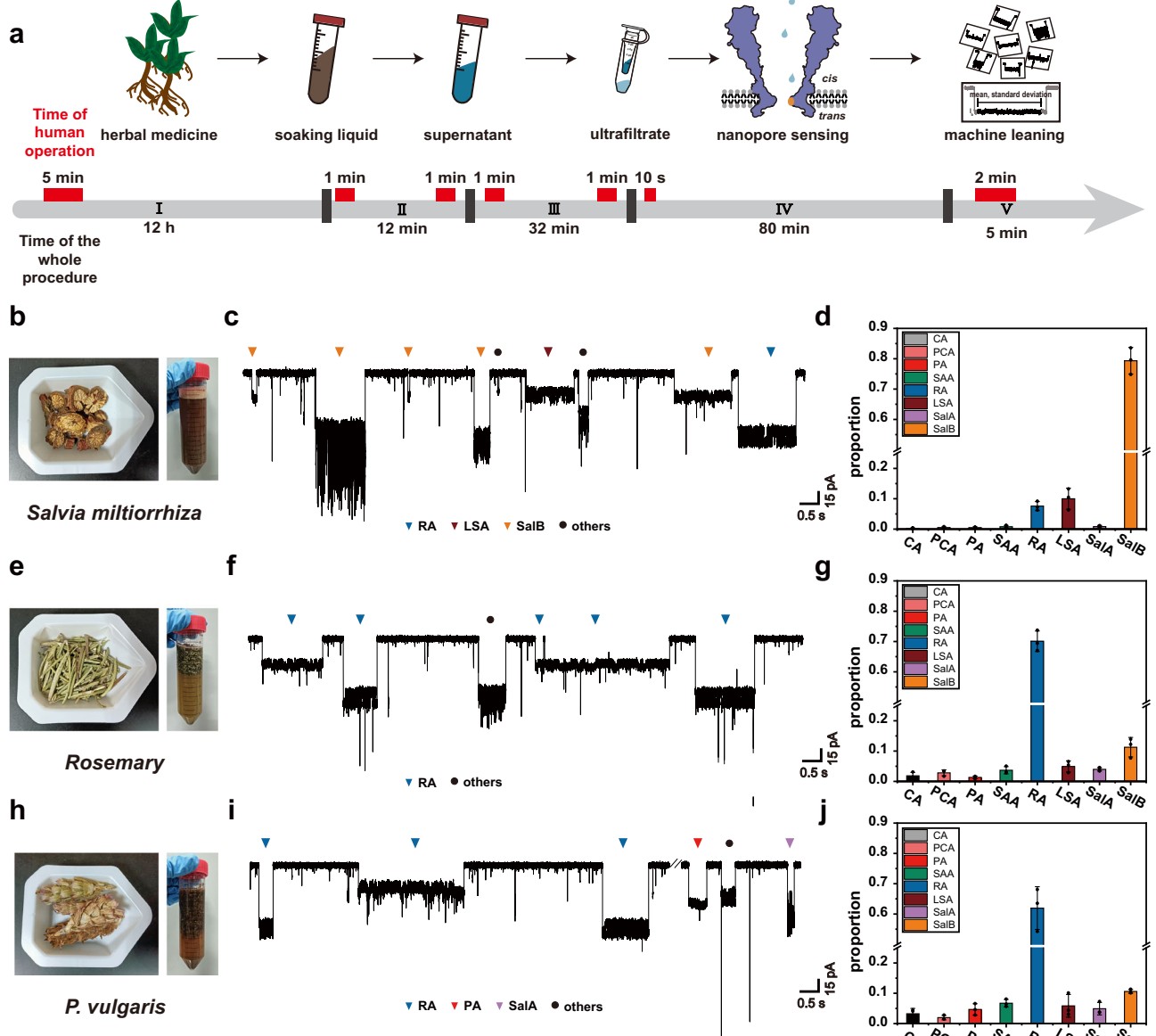

**Fig. 5 | Rapid identification of salvianolic acids in natural herbs. a** A workflow of nanopore identification of salvianolic acids directly from natural herbs. The gray timeline stands for the time of the whole procedure and red bars represent the time of human operation. Phase I: Sample pretreatment. Natural herbs were crushed and soaked in Milli-Q water for 12 h at 4 °C. Human operations: herb crushing and soaking (5 min). Phase II: Liquid collection. The soaking liquid was centrifuged at 4 °C and 1500 *g* for 10 min and the supernatant was collected. Human operations: centrifugation preparation (1 min) and supernatant collection (1 min). Phase III: Ultrafiltration. The collected supernatant was treated with a 3 kDa ultrafiltration tube at 4 °C and 1900 *g* for 30 min and the filtrate was collected. Human operations: ultrafiltration preparation (1 min) and filtrate collection (1 min). Phase IV: Nanopore sensing. 20 μL filtrate was added to the *cis* chamber of a nanopore device. Human operations: sample addition (10 s). Phase V: Data analysis. Human operations: automatic data analysis by machine learning (2 min). A more detailed workflow was also described in Methods and Supplementary Fig. 23. **b, e, h** Three types of

commercially available natural herbs including (**b**) *Salvia miltiorrhiza*, (**e**) *Rosemary* and (**h**) *P. vulgaris* and their corresponding soaking liquids. **c, f, i** Representative nanopore traces acquired with different herb samples. All events were identified by the trained KNN model and correspondingly labeled as RA (blue), LSA (wine-red), SalB (orange), PA (red), SalA (lavender) and others (black). The 'other' events represent events that don't belong to any previously trained salvianolic acid model compounds, based on results of outlier analysis (Supplementary Figs. 24, 26, 27). **d, g, j** The proportion of salvianolic acid events from results acquired with (**d**) *Salvia miltiorrhiza*, (**g**) *Rosemary* and (**j**) *P. vulgaris* (Supplementary Figs. 25, 28, 29). Data were presented as mean ± standard deviation values derived from results of three independent measurements (*N* = 3). The error bars represent standard deviation values. All above described results were acquired by nanopore measurement using MspA-90PBA in a buffer of 1.5 M KCl, 100 mM MOPS, pH 7.0 and a +100 mV bias, which was continually applied. Source data are provided as a Source Data file.

be ~ 12 ± 4 mg/g, which is highly consistent with those reported in previous literatures[48,73,74] (Supplementary Table 6).

The generality of rapid single molecule detection of salvianolic acids from natural herbs was further verified with *Rosmarinus offici-nalis L* (*Rosemary*) and *Prunella vulgaris* L (*P. vulgaris*). *Rosemary* is a kind of herbal medicine commonly used in natural additives[75] and for therapeutic purposes[76]. It has clinical effects of antioxidant[77],

anticancer[78] and anti-inflammatory[79] drugs. The therapeutic properties of *Rosemary* have been attributed to its phytochemical constituents[80], including salvianolic acids and terpenoids. An aqueous extract of *Rosemary* contains several kinds of salvianolic acids[80–84], among which RA is the most abundant component. As for *P. vulgaris*, the floral spikes of the plant has been generally applied to protect the liver[85], alleviate sore throats[86] and protect against breast cancer[87]. Previous

investigations have indicated that salvianolic acids are amongst the main bioactive components of *P. vulgaris*[88]. Judged by HPLC measurements[89–91], RA is the dominant salvianolic acid compound in extracts of *P. vulgaris*, but trace amounts of CA and PA were also detected.

As described in Fig. 5a, a volume of 20 μL ultrafiltration product from *Rosemary* and *P. vulgaris* were separately added to the *cis* chamber of the nanopore device to initiate the measurements. Representative raw traces acquired with these two herbal medicines were demonstrated (Fig. 5e, f, h, i). The interference events, which don't resemble to any events previously reported by the eight standard salvianolic acids, were removed by One-Class SVM (Supplementary Figs. 26, 27). Compared with standard analytes (Supplementary Table 2), nanopore measurements performed with natural herb extracts report more interference events. It is expected because a variety of *cis*-diols in natural herbs such as saccharides[36] and anthocyanin[92] may also bind to the PBA adapter to generate nanopore events. After removal of noise events (Supplementary Figs. 26, 27), all remaining events were predicted by the previously trained KNN model (Figs. 5f, i and Supplementary Movies 3, 4) and marked with color-coded dots in the corresponding scatter plots (Supplementary Figs. 28, 29). Three independent measurements were performed to show the measurement reproducibility. RA events were clearly identified from *Rosemary* and *P. vulgaris*. The calibration curve of RA, demonstrating a coefficient factor ($R^2 = 0.995$) is shown in Supplementary Fig. 30 and Supplementary Table 7. The corresponding $1/\tau_{on}$ values of RA events acquired with *Rosemary* and *P. vulgaris* were measured (Methods) and summarized in Supplementary Table 5. Afterwards, based on the calibration curve, the RA concentration in *Rosemary* and *P. vulgaris* were derived to be -1.26 ± 0.08 mg/g and -0.53 ± 0.12 mg/g (Supplementary Table 5), which are generally consistent with those previously investigated by HPLC[89,93–95] (Supplementary Table 6). The differences in RA concentration in our work and that reported in literatures may be due to the different material resource, sample pre-treatments and extraction processes.

According to the nanopore design, measurements performed at extremely high concentrations of analyte would result in the saturation of the PBA adapter and report inaccurate quantification. Thus, the effective concentration range for this measurement is defined to be the range of analyte concentration within which the $1/\tau_{on}$ is linearly correlated with the input analyte concentration (Supplementary Fig. 31, Supplementary Table 8). For measurements beyond the effective concentration range, sample enrichment or dilution will become necessary. However, all these nanopore analysis of natural herbs could not be performed using M2 MspA, which lacks a PBA adapter, again confirming the importance of the PBA adapter (Supplementary Fig. 32).

## Discussion

An engineered MspA nanopore was applied as a single molecule sensor of salvianolic acids. Eight salvianolic acids, including caffeic acid (CA), protocatechuic acid (PCA), protocatechualdehyde (PA), salvianic acid A (SAA), rosmarinic acid (RA), lithospermic acid (LSA), salvianolic acid A (SalA) and salvianolic acid B (SalB) were tested with this nanopore and their nanopore signatures are fully discriminable. Though all eight salvianolic acids have a wide range of spatial sizes, they can be simultaneously identified by the same nanopore. This high-resolution results from the sufficiently narrow pore constriction, which is comparable to target molecule size. The event features are primarily determined by interactions/local physio-chemical environment, leading to the extreme high resolution of the pore. To the best of our knowledge, nanopore sensing of salvianolic acids has never been previously reported. It is also observed that salvianolic acids containing multiple 1, 2-diol groups will report multiple types of binding events, demonstrating the superior resolution of MspA by its

distinguishing of different binding modes between salvianolic acids and the PBA adapter. A custom machine learning algorithm was also developed and a 99.0% accuracy was reported. The superior resolution of sensing and the high performance of machine learning enables recognition of salvianolic acid components directly from natural herbs such as *Salvia miltiorrhiza*, *Rosemary* and *P. vulgaris*. No chromatographic separation is necessary and the workflow of sensing only requires a few minutes of human operation. Though only demonstrated with salvianolic acids, the demonstrated principle should be generally suitable for other herbal medicines. Though only two event parameters, including $\Delta I/I_o$ and *S.D.* were used in the machine learning model building, the model performance is satisfying, suggesting that the raw data is of a high quality and data separation. In the future, when more analytes were to be simultaneously analyzed, more event parameters, such as skewness, kurtosis and dwell time may be further included. To further expand its sensing capacity, this hetero-octameric MspA may be installed with other reactive adapters, including those based on coordination chemistry[25,96,97], disulfide chemistry[98] or click chemistry[99,100], so that more diverse types of analytes may be sensed. With the increased complexity of the generated event features, the use of machine learning by simultaneous consideration of more event features[28] or deep learning[101] becomes indispensable. The whole setup may as well be further integrated into a miniaturized chip and used with a highly portable device, for applications in the field or in extreme situations when access to state-of-the-art instruments becomes impossible.

## Methods

### Preparation of a hetero-octameric *Mycobacterium smegmatis* porin A nanopore

All nanopore measurements were performed with MspA-90PBA, which is a hetero-octameric MspA specially engineered to contain an appended boronic acid at its pore constriction. In our previous works[28–30], it was also referred to as (N90C)₁(M2)₇. To prepare the hetero-octameric MspA, two genes respectively coding for M2 MspA-D16H16 and N90C MspA-H6 were simultaneously placed in a co-expression vector pETDuet-1 (GenScript). Briefly, the gene coding for N90C MspA-H6 was inserted between the restriction site of Nco I and Hind III. The gene coding for M2 MspA-D16H16 was inserted between the restriction site of Nde I and Blp I. The hexa-histidine tag (H6) added to the C-terminus of both genes serve to assist protein purification by nickel affinity chromatography. The sixteen aspartic acid tag (D16) added to the end of M2 MspA-D16H16 serves to generate a molecular weight difference between different assembly types of M2 MspA-D16H16 and N90C MspA-H6, which is critical in the purification of the target hetero-octameric MspA. The desired hetero-octameric MspA assembly, which contains a single unit of N90C MspA-H6 and seven units of M2 MspA-D16H6, is referred to as (N90C)₁(M2)₇ (Fig. 1a).

By heat shock transformation at 42 °C for 90 s followed with ice incubation, the constructed co-expression vector was transformed into *E. coli* BL21 (DE3) pLysS competent cells (Sangon Biotech). Then, a single colony was picked up and added to a LB broth with ampicillin (50 μg/mL) and chloramphenicol (34 μg/mL). The mixture was shaken at 37 °C and 175 rpm until OD600 = 0.6. Subsequently, IPTG was added to the LB broth with a final concentration of 0.1 mM and shaken for 24 h at 16 °C and 175 rpm for protein overexpression. The mixture was then centrifuged at 4 °C and 1500 *g* for 20 min to collect the bacterial pellet. The pellet was then resuspended in a 160 mL lysis buffer (100 mM Na2HPO4/NaH2PO4, 0.1 mM EDTA, 150 mM NaCl, 0.5% (v/v) Genapol X-80, pH 6.5) and heated at 60 °C for 50 min. After cooling to room temperature, the suspension was then centrifuged at 4 °C and 16000 g for 60 min and the supernatant was collected. The supernatant was filtered with a 0.2 μm syringe filter and then loaded to a HisTrapTMHP nickel ion affinity column (GE Healthcare) to obtain target protein MspAs. To further separate (N90C)₁(M2)₇ from other

pore assemblies, a 10% SDS-polyacrylamide gel was used to perform gel electrophoresis of the collected samples from nickel column purification. A + 160 V bias was continually applied for 16 h during the gel electrophoresis. Subsequently, the gel was stained with a coomassie brilliant blue solution (1.25 g coomassie brilliant blue R250, 225 mL MeOH, 50 mL glacial AcOH, 225 mL ultrapure water) for 4 h. Then, the de-staining solution (400 mL methanol, 100 mL glacial acetic acid, replenished with Milli-Q water to a volume of 1 L) was used for gel elution until the protein bands were clearly seen. The gel corresponding to the band of $(N90C)_1(M2)_7$ was then excised from the gel. The excised gel was crushed and immersed in an extraction solution (150 mM NaCl, 15 mM Tris-HCl, 0.2% (w/v) DDM, 0.5% (v/v) Genapol X-80, 5 mM TCEP, 10 mM EDTA, pH 7.5) for 12 h. The extracted $(N90C)_1(M2)_7$ was immediately used or stored at −80 °C for long-term use.

### 3-(maleimide) phenylboronic acid modification of $(N90C)_1(M2)_7$ MspA

To modify $(N90C)_1(M2)_7$ with a single phenylboronic acid, 5 µL $(N90C)_1(M2)_7$ and 2.5 µL 3-(maleimide) phenylboronic acid (500 mM, dissolved in DMSO) were mixed in a 43 µL buffer (1.5 M KCl, 100 mM MOPS, pH 7.0) for 10 min. The chemically modified $(N90C)_1(M2)_7$ was used during all nanopore measurements. The PBA modified $(N90C)_1(M2)_7$ is referred to as MspA-90PBA throughout the paper, if not otherwise stated.

### Nanopore measurements and data analysis

All electrophysiological measurements were performed with an Axopatch 200B patch-clamp amplifier paired with a Digidata 1550B digitizer. The custom measurement device was separated by a Teflon film contain a 100 µm diameter orifice. Prior to the measurement, the orifice was treated by 2% (v/v) hexadecane in pentane. During the measurements, each chamber of the device was first filled with a 500 µL buffer (1.5 M KCl, 100 mM MOPS, pH 7.0) and a pair of Ag/AgCl electrodes were inserted into the chambers, in contact with the buffers and electrically connected with the patch clamp amplifier to form a closed circuit. Conventionally, the chamber that is electrically grounded is defined as *cis* and its opposing chamber is defined as *trans*. Then a drop of 5 mg/mL DPhPC in pentane was added to each chamber for lipid bilayer formation. By pipetting the buffer up and down in either chamber, the lipid bilayer was spontaneously formed. Afterwards, biological nanopores were added to the *cis* chamber to trigger pore insertion. Until a single nanopore was inserted, the buffer in the *cis* chamber was manually exchanged to prevent further nanopore insertions.

All single-channel recordings were sampled at 25 kHz and low-pass filtered with a corner frequency of 1 kHz. This setting of data acquisition is suitable for nanopore events with a dwell time of -ms. A lower sampling rate is also advantageous to minimize the data size as well. If not otherwise stated, the +100 mV voltage was continually applied during all measurements and all measurements were carried out at room temperature (rt) (23 °C). All analytes were added to *cis*. All events were detected by Clampfit 10.7.

### Event feature extraction

For each raw trace, the start and the end time of each event was identified by Clampfit 10.7 and saved in csv files (Supplementary Fig. 14). The start and the end time were applied as the time stamps to segment an event from the raw trace. Events with a $t_{off} < 30$ ms were ignored. The segmented events were then used for further feature extraction using custom Python codes. The extracted features include relative blockage depth ($\Delta I/I_o$) and standard deviation ($S.D.$) and a feature matrix is formed. The mean current amplitude before the start and after the end of each event was calculated to derive the open pore current ($I_o$). Each relative blockage depth was derived according

to $\Delta I/I_o = (I_o - I_b)/I_o$, where $I_b$ represents the residual current of an event. After features extraction, the feature matrix results were saved as a csv file for all subsequent machine learning operations.

### Machine learning

Machine learning was performed in a Python environment. 500 events acquired with each type of analyte were collected and labeled to form a dataset. The dataset was randomly split into a training set (80% of the labeled data set) and a testing set (20%) for model training and model testing, respectively. The $\Delta I/I_o$ and $S.D.$ of events were employed as event features. The training set was standardized and then applied in the training using six common models, including K-NearestNeighbor (KNN), Extreme Gradient Boosting (Xgboost), Classification and Regression Tree (CART), Support Vector Machine (SVM), Gradient Boost Decision Tree (GBDT) and Random Forest (RF). According to the 10-fold cross validation results, KNN was selected as the optimum model and was applied for all further data analysis. The confusion matrix and the learning curve generated by KNN were employed for model evaluation. The trained model was saved and further applied for event predictions.

All machine learning models and training data generated in this study have been deposited in Figshare. Please follow the link for data download: https://figshare.com/s/3e3593adb4dfe4999068

### Composition quantification of herbal medicine

In the nanopore field, the reciprocal of the interevent interval ($1/\tau_{on}$) is widely known to be correlated with the target analyte concentration. It is also used for the quantitative measure of the analyte concentration[102,103]. The calibration curve of SalB was generated as the plot of $1/\tau_{on}$ versus the SalB concentrations (Supplementary Fig. 21, Supplementary Table 1). During nanopore sensing of the salvianolate injection, all events were predicted by the machine learning algorithm and the $t_{on}$ of SalB events were picked up for SalB quantification (Supplementary Fig. 20). By single-exponential fitting as described in Supplementary Fig. 2, the values of $1/\tau_{on}$ of SalB measured in the salvianolate injection was obtained (Supplementary Table 4). According to the calibration curve, the concentration of SalB in the salvianolate injection was calculated.

For natural herbs, the calculation of SalB content in *Salvia miltiorrhiza* and RA content in both *Rosemary* and *P. vulgaris* was identical to that previously demonstrated with the injection sample. The calibration curve of SalB and RA were generated as the plot of $1/\tau_{on}$ versus the SalB and RA concentrations (Supplementary Figs. 21, 30, Supplementary Tables 1 and 7). During nanopore sensing of natural herbs, all events were predicted by the machine learning algorithm after being treated with outlier analysis. The $t_{on}$ of SalB events in *Salvia miltiorrhiza* (Supplementary Fig. 25) and the $t_{on}$ of RA events in both *Rosemary* (Supplementary Fig. 28) and *P. vulgaris* (Supplementary Fig. 29) were picked up for quantification. By single-exponential fitting as described in Supplementary Fig. 2, the values of $1/\tau_{on}$ of SalB in *Salvia miltiorrhiza* and RA in both *Rosemary* and *P. vulgaris* measured in natural herbs were obtained (Supplementary Table 5). According to the calibration curve, the concentrations of SalB in *Salvia miltiorrhiza* and RA in both *Rosemary* and *P. vulgaris* were separately calculated.

### Pretreatments of natural herbs *Salvia miltiorrhiza, Rosemary* and *P. vulgaris*

Three natural herbs were identically pretreated as described in Supplementary Fig. 23. The commercially available natural herbs were first crushed into powder. 2 g herb powder was added to 40 mL of Milli-Q and soaked for 12 h at 4 °C. Subsequently, the soaking liquid was centrifuged for 10 minutes (4 °C, 1500 g) and then filtered with a 3 kDa ultrafiltration tube for 30 minutes (4 °C, 1900 g) to collect the filtrate. Finally, 20 µL filtrate of each natural herb was added to the nanopore device to initiate the measurement.

**Reporting summary**

Further information on research design is available in the Nature Portfolio Reporting Summary linked to this article.

## Data availability

The datasets supporting the findings of this study are available within the source data provided with this paper. All data used to generate the machine learning model have been deposited in Figshare. Please follow the link for data download: https://figshare.com/s/3e3593adb4dfe4999068 Source data are provided with this paper.

## Code availability

All custom machine learning models and training data were shared on Figshare. Please follow the link for data and code download: https://figshare.com/s/3e3593adb4dfe4999068

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

## Acknowledgements

This project was funded by National Natural Science Foundation of China (Grant No. 22225405, No. 31972917, to S.H.), the National Key R&D Program of China (Grant No. 2022YFA1304602, to S.H.), the Fundamental Research Funds for the Central Universities (Grant No.020514380257, to S.H.), Programs for high-level entrepreneurial and innovative talents introduction of Jiangsu Province (individual and group program, to S.H.), Natural Science Foundation of Jiangsu Province (Grant No. BK20200009, to S.H.), Excellent Research Program of Nanjing University (Grant No. ZYJH004, to S.H.).

## Author contributions

S.H., P.P.F., and S.Y.Z. conceived the project. P.P.F. and S.Y.Z. prepared the MspA nanopores. P.P.F., S.Y.Z., Y.Q.W., T.L., H.H.Z. performed the measurements. P.P.F. designed the machine-learning algorithms. P.K.Z. set up the instruments. S.H. and P.P.F. wrote the paper. S.H. supervised the project.

## Competing interests

S.H., S.Y.Z., and Y.Q.W. have filed patents describing the preparation of heterogeneous MspA and its applications thereof. The remaining authors declare no competing interests.
