## [Peer Review File · Nature Communications]

REVIEWER COMMENTS

Reviewer #1 (Remarks to the Author):

The detection of multiple salvianolic acids at single-molecule level in herbal medicines has been achieved through the combination of mutant bionanopore applications with artificial intelligence (AI). This detection method is based on intermolecular interactions between the inner wall of bionanopores and two hydroxyl groups present in salvianolic acids. By leveraging these interactions to distinguish multiple salvianolic acids, this method showcases the exceptional versatility of bionanopores as single-molecule sensors. Furthermore, the successful analysis of herbal medicine extracts through this newly developed nanopore system demonstrates its potential as a multimolecular analytical tool. Although this paper addresses several challenges related to the development of bionanopores as analytical tools, several issues remain.

1. The combination of nanopore applications and machine learning (ML) has been employed in DNA sequencers. Further, the combination of nanopore applications and ML has recently seen rapid advancement. Accordingly, the authors should cite relevant representative references to substantiate the assertions made regarding ML in the relevant section.
2. The authors justify the introduction of ML as a means to avoid bias resulting from human judgment. However, notably, the two features established herein were determined by the authors, which is clearly a form of bias. Accordingly, the abovementioned justification is inappropriate.
3. Figure 2i demonstrates the effective discrimination of eight acids through statistical means. This implies that identification is feasible without introducing ML. Then why is ML necessary? This question is inherently tied to the response to the point discussed above.
4. The authors should elucidate why $1/\tau$ is proportional to concentration.
5. Figures 4 and 5 are intended to show that the combination of bionanopore applications and AI may quantitatively analyze eight specific acids found in herbal medicines. However, the combination does not provide a clear evidence for such possibility. However, for the analysis, a sample containing eight acids with known concentrations and mixing ratios could be prepared. Subsequently, the predetermined ratios and those calculated from the combination can be compared to verify the effectiveness of the combination system.
6. Why is a 25-KHz sampling frequency and a 1-kHz low-pass filter used? The use of a lower than usual time resolution for nanopore measurements may be attributed to the extended dwell time of acid molecules due to intermolecular interactions.
7. Generally, in studies employing ML, raw measurement data for analysis are shared publicly among researchers. Similarly, the raw measurement data used in ML elements of this study should be publicly available.

Reviewer #2 (Remarks to the Author):

In this study, the authors have reported a proof-of-concept study for the identification of bioactive compounds based on salvianolic acid, including caffeic acid, protocatechuic acid, protocatechualdehyde, salvianolic acid A, rosmarinic acid, lithospermic acid, salvianolic acid A and salvianolic acid B by using a phenylboronic acid appended Mycobacterium smegmatis porin A (MspA) nanopore. The machine learning (ML) classification method is adopted for the classification of eight bioactive molecules. The study is interesting and can be published after a revision in consideration of the following comments.

Major Comments:

1. The current abstract suffers from poor writing issues and a lack of coherence between sentences, potentially limiting its ability to engage a wider audience of Nature Communications. “The manufacturing and the use of herbal medicines lack reliable standards” What is this sentence to do with nanopore detection? Write the full form of *P. vulgaris*.
2. The eight bioactive molecules have a wide range of spatial sizes. How the nanopore size is fixed. For small molecules, the high influx of ionic current signal may wash away the molecule events. What kind of interaction do the molecules show with the phenylboronic acid (PBA) adaptor of MspA-90PBA to produce nanopore events? What does the author mean by “cis-diol structure should react with the PBA”. If it's a chemical reaction, then it may lead to pore-clogging issues.
3. CA, PCA, and PA molecules are of comparable size with a diol group, which is likely to interact similarly with the nanopore adaptor. So, how are they identified on the basis of their ionic current blockage events?
4. The authors have demonstrated the identification of cis-diols directly from natural herb extracts by MspA-90PBA nanopore. Can the authors comment on the capability of the nanopore toward the identification of trans-diols? If there are any reports available for the identification of trans-diols, the authors should cite those with key highlights.
5. Can the authors comment on why they have chosen 80:20 particularly as a train-test split?
6. The machine learning classification algorithm SVM depends on two important parameters C and γ that control the quality of the result. The authors may check the results with a possible combination of C and γ . A color map for the parameter space and accuracy would be helpful (ACS Appl. Mater. Interfaces 2019, 11, 20, 18494–18503).
7. Can the authors comment on the underlying mechanism of current variation for sensing salvianolic acids through the MspA nanopore?
8. Regarding data sets, each salvianolic acid is showing three types of events. How the signature signal of bioactive compounds is evaluated during database preparation. Why only two features other than ionic current blockage (I_0) are considered? Other than $\Delta I/I_0$ and S.D, what will be the effect of other mathematical operations ($\log I$, I_{min}/I , I_{mx}/I etc.) on the ML classification results?
9. What is role of concentration of bioactive molecules on the ML classification, especially for CA, PCA, and PA molecules.

10. The author argued, "To avoid human judgemental bias," the ML method is applied. How is ML classification assisting or accelerating bioactive molecule identifications? Since data cleaning, denoising, and feature extraction are complex processes, any slight aberration may mislead the prediction.

11. What could be the reason behind the best performance of simpler KNN models than RF and XGBoost models with these data sets? What are the hyperparameters considered? Can the optimized ML models be integrated with the experimental setup for automatic identification of molecules. How feasible and efficient it would be?

12. Can author comment on which feature is mainly driving the better classification results?

Minor Comments:

1. As the methods section is given in the main manuscript, the authors should mention it as methods only not as SI methods as given in line no.88.
2. Figure citations are not in order. Should Fig 1b be cited before Fig 1a?
3. Throughout writing of the manuscript can be improved.

Author Response Letter

Reviewer #1 (Remarks to the Author):

Review Comments:

The detection of multiple salvianolic acids at single-molecule level in herbal medicines has been achieved through the combination of mutant bionanopore applications with artificial intelligence (AI). This detection method is based on intermolecular interactions between the inner wall of bionanopores and two hydroxyl groups present in salvianolic acids. By leveraging these interactions to distinguish multiple salvianolic acids, this method showcases the exceptional versatility of bionanopores as single-molecule sensors. Furthermore, the successful analysis of herbal medicine extracts through this newly developed nanopore system demonstrates its potential as a multimolecular analytical tool. Although this paper addresses several challenges related to the development of bionanopores as analytical tools, several issues remain.

Author Response:

We sincerely acknowledge reviewer 1 for an accurate description of the main topic of this work and your recognition of the potential and the exceptional versatility of this newly developed nanopore system for complex herbal medicine analysis. We also thank reviewer 1 for considering our analysis of herbal medicine extracts to be successful. We are grateful for the positive comments from reviewer 1 describing that this nanopore system is “potential as a multimolecular analytical tool”. We also sincerely acknowledge reviewer 1 for your other valuable comments on this manuscript. Detailed responses to your comments are listed below in a point by point style for the ease of your reference.

Review Comments:

The combination of nanopore applications and machine learning (ML) has been employed in DNA sequencers. Further, the combination of nanopore applications and ML has recently seen rapid advancement. Accordingly, the authors should cite relevant representative references to substantiate the assertions made regarding ML in the relevant section.

Author Response:

We thank reviewer 1 for this valuable suggestion on the reference. To fully address this reference issue, we have added relevant literatures about machine learning assisted

nanopore applications in the revised manuscript (now ref. 28 and 30). Some relevant discussions were also added as well. The added discussion is also pasted here, for the ease of your reference.

“In the field of nanopore research, machine learning has been widely applied to assist data analysis of nucleic acid sequencing and single molecule sensing.”

Review Comments:

The authors justify the introduction of ML as a means to avoid bias resulting from human judgment. However, notably, the two features established herein were determined by the authors, which is clearly a form of bias. Accordingly, the abovementioned justification is inappropriate.

Author Response:

We sincerely acknowledge reviewer 1 for raising this insightful discussion. We also apologize that our description of machine learning may have misled you. What we really wanted to express is that no matter which event parameters were selected, whenever the machine learning algorithm is established, it is identically used without any further human interferences and a perfect judgement consistency is guaranteed. The same algorithm could also be used for data acquired by different groups so that the performance of data accuracy could be fairly compared. Besides, the ML algorithm also serves to save the effort of human judgement.

The reviewer is correct that we, the researchers, have decided to use these two event features during model building. However, in the nanopore field, it is a common practice to use $\Delta I/I_0$ and S.D. of the blockage levels as event features for nanopore event discriminations (*ACS Nano* 16, 7258-7268 (2022). <https://doi.org/10.1021/acsnano.1c11455>; *Angew. Chem. Int. Ed.* 61, e202209970 (2022). <https://doi.org/10.1002/anie.202209970>). Specifically in this case, since the difference in these two event features are so clear and the performance of the generated ML algorithm is also working great, we didn't attempt to try other event features in this ML algorithm.

Here, the model performance was characterized by confusion matrix and learning curve results (**Figure 3c** and **Figure S15**), according to which it is confirmed that the model constructed by the two parameters has great generalization and is neither overfitting nor underfitting and the model accuracy is sufficiently high. In the future, when more types of analytes are included in the same system, more event features may be needed to further improve the model performance.

We acknowledge reviewer 1 again for your comment on the selection of the event features. To fully address this issue, relevant discussions have been added to the revised

manuscript. For the ease of your reference, they are also pasted below as well.

“In this work, the data acquired with complex samples leads to the difficulty of events identification by human eyes. Besides, when large amount of data is involved, event identification automation also becomes urgent. To also quantitatively assist event identification and sensing performance evaluation, a Python-based machine learning algorithm was developed.”

“Though only two event parameters, including $\Delta I/I_0$ and $S.D.$ were used in the machine learning model building, the model performance is satisfying, suggesting that the raw data is of a high quality and data separation. In the future, when more analytes were to be simultaneously analyzed, more event parameters, such as skewness, kurtosis and dwell time may be further included.”

Review Comments:

Figure 2i demonstrates the effective discrimination of eight acids through statistical means. This implies that identification is feasible without introducing ML. Then why is ML necessary? This question is inherently tied to the response to the point discussed above.

Author Response:

We acknowledge reviewer 1 for the recognition of the discrimination power of MspA-90PBA in simultaneous discrimination of eight salvianolic acids. The reviewer is correct, solely by the results presented in the scatter plot, all eight salvianolic acids were already discriminated and a high quality of data is seen. We however find the introduction of the ML algorithm more advantageous than visual inspection in these aspects of view.

- 1. The automation.** The whole process of machine learning assisted event identification, including feature extraction, model training and model prediction et al, can be automatically performed with the custom Python-based algorithms. This is extremely useful when a large quantity of data was generated. However, it is extremely time-consuming and labor-intensive for humans to do that manually and visually.
- 2. The speed of the analysis.** Specifically, for herbal medicine analysis, the whole automated data analysis can be finished in just ~min using machine learning algorithms, which is not easily achieved solely by human judgment.
- 3. Further introduction of multiple event features.** In this work, we are satisfied with the performance achieved by simultaneously considering two event features, the relative blockage depth ($\Delta I/I_0$) and the standard deviation ($S.D.$). However, this

is only for simultaneous discrimination of eight salvianolic acids. Soon, when more analytes were further introduced to this system, more events parameters including, skewness (skew), kurtosis (kurt), dwell time and many others may be further included to fit the sensing need. The establishment of the current ML algorithm is technically prepared for that.

- 4. A quantitative and objective judgment of the discrimination accuracy.** The machine learning algorithm provides a systematic, quantitative and objective measurement of the data discrimination performance. These are based on results of the cross-validation accuracy, confusion matrix and the learning curves. However, it is hard to draw quantitative conclusion only by visual judgement of human.
- 5. The preparation for noise background events.** For complex natural samples, such as real natural herbs (**Figure S24**), many interfering events can seriously affect visual judgment, which is hard for humans to judge. The introduction of machine learning algorithms can automatically remove interfering events to guarantee the robustness of data analysis against noise background events.

Finally, we again thank reviewer 1 for raising this insightful discussion of why a machine learning algorithm is needed when visual judgement by human appears to be all right. To fully address this issue, some relevant discussions have been added to the revised manuscript. These added discussions are also pasted here for the ease of your reference.

“In this work, the data acquired with complex samples leads to the difficulty of events identification by human eyes. Besides, when large amount of data is involved, event identification automation also becomes urgent. To also quantitatively assist event identification and sensing performance evaluation, a Python-based machine learning algorithm was developed.”

Review Comments:

The authors should elucidate why $1/\tau$ is proportional to concentration.

Author Response:

We acknowledge reviewer 1 for this constructive comment. We also apology that we didn't explain this concept with enough details, in the original manuscript. Here, in the nanopore field, it is a commonly accepted conclusion that the $1/\tau_{on}$ is proportional to the analyte concentration. This is also widely used in the quantification of analyte during nanopore readout. This was both reported in our previous works (*Angew. Chem. Int. Ed.* 61, e202203769 (2022). <https://doi.org/10.1002/anie.202203769>; *Chem. Eur. J.* 28, e202201033 (2022). <https://doi.org/10.1002/chem.202201033>; *Nat. Commun.* 12, 5811 (2021). <https://doi.org/10.1038/s41467-021-26054-9>.) and also reported in other

literatures (*J. Am. Chem. Soc.* 131, 3772-3778 (2009). <https://doi.org/10.1021/ja809663f>; *Langmuir* 27, 19-24 (2011). <https://doi.org/10.1021/la104264f>).

To explain this with more details, the process of boric acid ester formation and dissociation occurring inside the nanopore can be simplified as a system of bimolecular reactions. The bimolecular model can be described by the following kinetic scheme:

Here, **P** stands for the PBA adapter in the nanopore. **A** refers to the analyte molecules. **P · A** stands for the boric acid ester formed in a nanopore. k_{on} is the association rate constant and k_{off} is the dissociation rate constant. Obviously, the above bimolecular reaction demonstrated the fact that there are two reactant molecules (PBA and 1,2-diols) involved in the reaction and the reaction rate will be also affected by the concentration of these two molecules. Therefore, according to the simple bimolecular reaction kinetic equation (*Nature* 398, 686–690 (1999). <https://doi.org/10.1038/19491>; *Phil. Trans. R. Soc. Lond. B* 300, 1– 59 (1982). <https://doi.org/10.1098/rstb.1982.0156>), the association rate constant k_{on} and dissociation rate constant k_{off} of 1,2-diols can be obtained respectively:

$$1/\tau_{on} = k_{on}[\text{A}]$$

$$1/\tau_{off} = k_{off}$$

in which **[A]** is the concentration of the 1,2-diols. According to the above equations, the reason that $1/\tau_{on}$ is proportional to concentration of 1,2-diols was clearly illustrated, which is consistent with the results showed in now **Figure S3**.

Eventually, we again thank reviewer 1 for this comment. We believe that the above provided discussions should have well addressed your inquiry. A simple discussion has also been added to the revised manuscript. It is also pasted below for the ease of your reference. Detailed derivation of related equations can also be found in this reference (*Nature* 398, 686–690 (1999). <https://doi.org/10.1038/19491>).

“In the nanopore field, the reciprocal of the interevent interval ($1/\tau_{on}$) is widely known to be correlated with the target analyte concentration. It is also used for the quantitative measure of the analyte concentration”

Review Comments:

Figures 4 and 5 are intended to show that the combination of bionanopore applications and AI may quantitatively analyze eight specific acids found in herbal medicines. However, the combination does not provide a clear evidence for such possibility.

However, for the analysis, a sample containing eight acids with known concentrations and mixing ratios could be prepared. Subsequently, the predetermined ratios and those calculated from the combination can be compared to verify the effectiveness of the combination system.

Author Response:

We thank reviewer 1 for raising this constructive discussion about quantitative analysis of herbal medicines. To fully address your inquiry, a mixture with predetermined concentrations of specific acids found in herbal medicines is measured and quantitatively analyzed as demonstrated in **Figure RL 1.1**. In principle, the concentrations of all salvianolic acids can be calculated by their corresponding calibration curves as described in **Methods**. Here, taking SalB as an example (now **Figure S22** in the revised manuscript), the concentration of SalB calculated from the combination of nanopore measurements and machine learning is derived to be $\sim 23.6 \mu\text{M}$, which is highly consistent with the actual value in the mixture, $\sim 25 \mu\text{M}$. Thus, the capacity of nanopore in the quantification of bioactive compounds in herbal medicines are experimentally approved and your inquiry should have been fully resolved. Eventually, we again thank reviewer 1 for your comment on the mentioned discussion. Some relevant discussions and figure have been also added to the revised manuscript to complete this discussion, as pasted below.

“To verify the effectiveness of the quantitative results, a mixture with predetermined concentrations of eight salvianolic acids was measured and quantitatively analyzed as described in Figure S22. In principle, the concentrations of all salvianolic acids can be calculated by their corresponding calibration curves as described in Methods. Here, taking SalB as an example (Figure S22), the concentration of SalB in the mixture was derived to be $\sim 23.6 \mu\text{M}$, highly consistent with actual value, $\sim 25 \mu\text{M}$, illustrating the capacity of nanopore in the quantification of bioactive compounds in herbal medicines.”

Figure RL 1.1. Quantitative analysis of eight salvianolic acids simultaneous sensing. (a) The event scatter plot of $\Delta I/I_0$ versus $S.D.$ of results acquired with a mixture of eight salvianolic acids. A total of 794 events were included ($n = 794$) and all event identities were predicted by machine learning. All analytes were added to *cis* to reach the desired final concentrations and a +100 mV bias was continually applied. Specifically, the final concentration of CA and SAA was $10 \mu\text{M}$, that of PCA was $20 \mu\text{M}$, that of PA, RA, LSA and SalA was $5 \mu\text{M}$ and that of SalB was $25 \mu\text{M}$. Events in the scatter plot were from a 40 min continually recorded trace. (b) A quantitative demonstration of

identified salvianolic acid events from the mixture. (c) The histogram of t_{on} acquired from SalB events. By single-exponential fitting as described in **Figure S2**, the values of $1/\tau_{on}$ of SalB measured in the mixture was obtained and a coefficient factor (R^2) was shown. Subsequently, the concentration of SalB in the mixture can be derived as described in **Methods** and **Figure S21**.

Review Comments:

Why is a 25-KHz sampling frequency and a 1-kHz low-pass filter used? The use of a lower than usual time resolution for nanopore measurements may be attributed to the extended dwell time of acid molecules due to intermolecular interactions.

Author Response:

We sincerely thank reviewer 1 for raising this discussion and agree with reviewer 1. First of all, we would like to clarify that all our demonstrated nanopore herbal medicine measurement can be carried out at other sampling frequency and low-pass filtering conditions. However, a high sampling frequency may generate too high noises and large data size. Whereas, a low sampling frequency may distort the data quality. With the PBA adapter placed in the pore constriction, the nanopore events of salvianolic acids are significantly extended so that the measurement can be carried out with a lower sampling frequency and a lower low-pass filter. Still, we apology that we didn't include discussions of the selection of the sampling frequency and the low-pass filter. Some relevant discussions have been added to the revised manuscript and this issue should have been fully resolved.

“This setting of data acquisition is suitable for nanopore events with a dwell time of ~ms. A lower sampling rate is also advantageous to minimize the data size as well.”

Review Comments:

Generally, in studies employing ML, raw measurement data for analysis are shared publicly among researchers. Similarly, the raw measurement data used in ML elements of this study should be publicly available.

Author Response:

We sincerely acknowledge reviewer 1 for providing this suggestion. The related machine learning models and core data are shared in Figshare. The link to the code and models is also provided here: <https://figshare.com/s/3e3593adb4dfe4999068>

Eventually, we again acknowledge reviewer 1 for raising so many great suggestions for our manuscript. These suggestions have all been fully addressed and the overall quality of the manuscript is further improved.

Reviewer #2:

Review Comments:

In this study, the authors have reported a proof-of-concept study for the identification of bioactive compounds based on salvianolic acid, including caffeic acid, protocatechuic acid, protocatechualdehyde, salvianolic acid A, rosmarinic acid, lithospermic acid, salvianolic acid A and salvianolic acid B by using a phenylboronic acid appended Mycobacterium smegmatis porin A (MspA) nanopore. The machine learning (ML) classification method is adopted for the classification of eight bioactive molecules. The study is interesting and can be published after a revision in consideration of the following comments.

Author Response:

We highly acknowledge reviewer 2 for a precise description of the main topic of this manuscript. Besides, we also sincerely acknowledge reviewer 2 for your recommendation to the publication of this article. We also treasure your valuable comments aiming to improve the quality of this manuscript. These comments have been addressed below in a point by point style for the ease of your reference.

Review Comments:

The current abstract suffers from poor writing issues and a lack of coherence between sentences, potentially limiting its ability to engage a wider audience of Nature Communications. “The manufacturing and the use of herbal medicines lack reliable standards” What is this sentence to do with nanopore detection? Write the full form of *P. vulgaris*.

Author Response:

We acknowledge reviewer 2 for your suggestion with the writing of the abstract. We sincerely apology for causing the confusion to reviewer 2. In the revised manuscript, the relevant abstract has been revised and the full form of *P. vulgaris*., *Prunella vulgaris*, has also been included. We again thank reviewer 1 for your kind suggestion.

Review Comments:

The eight bioactive molecules have a wide range of spatial sizes. How the nanopore size is fixed.

Author Response:

We thank reviewer 2 for raising this inspiring question. The reviewer is correct that the

eight mentioned bioactive molecules have a wide range of spatial sizes. However, this dynamic range of analyte size is not a problem but a benefit for sensing. Because analytes of different sizes generally maximize the differences in their nanopore events. Besides, these eight bioactive molecules may simultaneously present in the same set of sample, so it is also better to perform all the measurement using the same pore sensor.

Based on the design of this pore sensor, a PBA adapter was placed at the pore constriction. As long as the cis-diol moiety of the analyte can reach the pore constriction to react with the PBA adapter, the sensing could be achieved. **A wider range of analyte size may even help to produce more distinguishable nanopore events.** We chose MspA because it has an excellent resolution and is suitable to thoroughly amplify subtle differences between all eight salvianolic acids. According to the results shown in **Figure 2**, we believe that it is doing a good job.

Eventually, to fully address your inquiry, based on above explanations, some relevant discussions of a nanopore with a fixed aperture size was selected have been added in the revised manuscript. It is also pasted below for the ease of your reference.

“Though all eight salvianolic acids have a wide range of spatial sizes, they can be simultaneously identified by the same nanopore, acknowledging the high resolution of this MspA sensor.”

Review Comments:

For small molecules, the high influx of ionic current signal may wash away the molecule events. What kind of interaction do the molecules show with the phenylboronic acid (PBA) adaptor of MspA-90PBA to produce nanopore events?

Author Response:

We acknowledge reviewer 2 for providing this valuable discussion. The reviewer is correct, the small molecule analytes will be washed away, appearing as a transient nanopore events when monitored during single channel recording. To overcome this issue, a reversible chemical interaction must be established between the pore and the analyte. Specially in this case, a reversible covalent chemical reaction is established between the PBA adapter on the pore and the cis-diol moiety of the analyte. Consequently, nanopore events of salvianolic acids were clearly observed. However, with the same pore but without the PBA adapter, no well defined nanopore events could be observed with salvianolic acids at all (**Figure S26**).

We again sincerely acknowledge this valuable inquiry from reviewer 2. To fully address this issue, some relevant discussions of the pore design principle were added to the revised manuscript.

“The PBA adapter also serves to reversibly capture and release the analyte during recording, so that an extended event dwell time is reported.”

Review Comments:

What does the author mean by “cis-diol structure should react with the PBA”. If It’s a chemical reaction, then it may lead to pore-clogging issues.

Author Response:

We thank reviewer 2 for this comment. According to the previous literatures, it is generally known that the cis-diol structure can react with PBA. This reaction principle has been widely used in self-healing material (*Adv. Mater.* 28, 86-91 (2016). <https://doi.org/10.1002/adma.201502902>), drug delivery (*Acc. Chem. Res.* 52, 3108–3119 (2019). <https://doi.org/10.1021/acs.accounts.9b00292>) et al. In an aqueous environment, the reaction of cis-diols and PBA is highly reversible. The rate of this reaction could also be modulated by the buffer pH as well (*ACS Nano* 2023, 17, 3, 2881–2892. <https://doi.org/10.1021/acsnano.2c11286>).

Binding and dissociation of cis-diols to the PBA adapter could also be probed in single molecule during single channel recording. Acknowledging this reversible interaction, binding of salvianolic acid to the pore lumen can last for ~ ms, which is suitable for nanopore sensing. **If this interaction between the pore and the analyte is too strong or irreversible, then pore clogging could happen. However, with the experiment condition we have demonstrated, this has never been observed at all.** Still, some discussions on this topic have been added to the revised manuscript to fully address this issue. Your inquiry is highly acknowledged.

“The PBA adapter also serves to reversibly capture and release the salvianolic acid analyte during recording, so that an extended event dwell time is reported.”

Review Comments:

CA, PCA, and PA molecules are of comparable size with a diol group, which is likely to interact similarly with the nanopore adaptor. So, how are they identified on the basis of their ionic current blockage events?

Author Response:

We acknowledge reviewer 2 for raising this insightful discussion and for the recognition of discrimination of different salvianolic acids with high resolution. The reviewer is correct that CA, PCA and PA are about the same size. However, their chemical structures are still different. Acknowledging the high resolution of MspA, this minor difference can still be well resolved so that highly distinguishable nanopore events are

produced. In some cases, even epimers (structurally different but with exactly the same size and molecular weight) can be distinguished by nanopore as well (*J. Am. Chem. Soc.* 2022, 144, 30, 13717–13728). The interactions between the analyte and the internal pore lumen may also produce noise signals for different analytes. These may also help in the discrimination of structurally similar analytes.

Eventually, we appreciate reviewer 2 for raising this inspiring discussion which may lead to the discussion of the high resolution of nanopore sensors. Some previous references on this topic and relevant discussions are added to the revised manuscript to fully resolve this issue.

“When captured by the PBA adapter and chemically confined in the pore lumen, the analyte may further interact with the amino acid residues of the pore to produce characteristic noises on top of the blockage levels, which is useful for event identification. Salvianolic acids containing multiple 1, 2-diol structures also report multiple event types, and these event types are also distinguishable in the corresponding scatter plot of $\Delta I/I_0$ versus S.D., acknowledging the high resolution of this engineered MspA sensor.”

Review Comments:

The authors have demonstrated the identification of cis-diols directly from natural herb extracts by MspA-90PBA nanopore. Can the authors comment on the capability of the nanopore toward the identification of trans-diols? If there are any reports available for the identification of trans-diols, the authors should cite those with key highlights.

Author Response:

We thank reviewer 2 for raising this suggestion. According to relevant literatures, the PBA can reversibly react with cis-diols in a nanopore (*Angew. Chem. Int. Ed.* 61, e202203769 (2022). <https://doi.org/10.1002/anie.202203769>). However, the reaction of PBA with trans-diols in a nanopore has not been reported yet. We assume that the trans-diol should fail to report any well-defined events when probed by this sensor.

To experimentally demonstrate that, trans-1,2-cyclohexanediol is taken as an example of trans-diols for nanopore sensing with MspA-PBA. As **Figure RL 2.1** (now **Figure S1** in the revised manuscript) shown, only extremely short residing events with no well-defined event features were shown, again confirming that trans-diols would fail to report any useful events when probed by this nanopore sensor.

To fully address this inquiry, some relevant discussions have also been added to the revised manuscript as well, as pasted below.

“One drawback is that the PBA adapter is in principle not suitable for trans-diols.”

Figure RL 2.1. The single molecular sensing of the trans-1,2-cyclohexanediol. **Left:** The chemical structure of trans-1,2-cyclohexanediol. **Right:** The trans-1,2-cyclohexanediol sensing at different concentrations using MspA-PBA.

Review Comments:

Can the authors comment on why they have chosen 80:20 particularly as a train-test split?

Author Response:

We acknowledge reviewer 2 for this valuable comment. First of all, this train-test split of 80:20 was widely applied in other machine learning tasks and nanopore works (*Nat. Biotechnol.* 39, 336–346 (2021). <https://doi.org/10.1038/s41587-020-0712-z>; *Nat. Nanotechnol.* 17, 976–983 (2022). <https://doi.org/10.1038/s41565-022-01169-2>). Sometimes, a 90:10 train-test split (*Nat. Commun.* 12, 3726 (2021). <https://doi.org/10.1038/s41467-021-24001-2>) or a 75:25 train-test split (*Proc. Natl. Acad. Sci.* 118, e2022806118, (2021). <https://doi.org/10.1073/pnas.2022806118>) can also be used.

The basic principle of train-test split is that the test samples are not included in the train samples, so that the generalization of the model could be estimated. We are generally satisfied with the performance of the established machine learning model. Thus, we didn't attempt to optimize it with different train-test split ratios. Still, some relevant references have been added in the revised manuscript to support this part of discussion.

Review Comments:

The machine learning classification algorithm SVM depends on two important parameters C and γ that control the quality of the result. The authors may check the results with a possible combination of C and γ . A color map for the parameter space and accuracy would be helpful (*ACS Appl. Mater. Interfaces* 2019, 11, 20, 18494–18503).

Author Response:

We thank reviewer 2 for raising such insightful discussion on the parameter optimization during machine learning. Reviewer 2 is correct that the C and γ parameters in the SVM model could be further optimized to achieve a better performance of sensing. However, specifically for this case, the event features of all eight salvianolic acids are already so different and a high classification performance (99.0%) is already achieved with the default setting of C and γ . Further tuning of C and γ thus becomes not urgent.

Still, to verify this, the two parameters of SVM model were varied between every 10^{-5} and 10^5 . For each of combination of C and γ , SVM model was evaluated using 10-fold cross validation and the corresponding accuracy score is reported. With the grid search, the parameter space and corresponding accuracy can be encoded to a color map as shown in **Figure RL 2.2**. The combination of C and γ that reports the highest mean accuracy of 99.0% was determined, which corresponds to $C = 100$ and $\gamma = 1$. The default parameter setting is $C = 1$ and $\gamma = 0.5$, which reported an accuracy of 98.7% (**Figure 3b**). With this demonstration, it is quite clear that improvement gained from further tuning of the parameters is insignificant, consistent with our expectation.

Eventually, we would like to thank reviewer 2 again for raising this insightful suggestion of parameter tuning. Some relevant discussions were added to the revised manuscript to fully address this issue as well.

“All above trainings were carried out with the default hyperparameter settings.”

Figure RL 2.2. The parameter optimization of SVM model for eight salvianolic acids mixture prediction. (a) The color map showing the tuning of C and γ . The prediction results of eight salvianolic acids simultaneous sensing obtained by the (b) optimized and (c) default SVM models.

Review Comments:

Can the authors comment on the underlying mechanism of current variation for sensing salvianolic acids through the MspA nanopore?

Author Response:

We acknowledge reviewer 2 for the comment of the underlying mechanism of event

acquisition. First of all, all salvianolic acids tested in this paper have different chemical structures, meaning that their nanopore events should be different. However, to thoroughly amplify this difference, a pore with a sufficiently high resolution would be needed. The MspA-PBA is just a pore of this kind. It has an overall conical pore lumen, which efficiently focuses the ionic current to the narrowest spot of the pore constriction. Thus, the events generated by different salvianolic acids are thoroughly discriminated.

For the analyte, when captured by the PBA adapter and confined in a narrow pore constriction, it may further interact with the amino acid residues in the internal pore lumen, producing fluctuations on top of the pore blockage level. These fluctuations, which appear as differences in the SD of the blockage current, further helps in the discrimination of different salvianolic acids. However, since there are no experiment methods that can probe the exact single molecule analyte configuration in the pore constriction during single channel recording, it is almost impossible to directly correlate the molecular configuration of the analyte and the corresponding nanopore readout. However, some molecular dynamics study may provide some inspiring hints.

Thus, to fully address this issue, I think a relevant discussion could be added to the revised manuscript, as pasted below. Your inspiring inquiry is again highly evaluated.

“When captured by the PBA adapter and chemically confined in the pore lumen, the analyte may further interact with the amino acid residues of the pore to produce characteristic noises on top of the blockage levels, which is useful for event identification.”

Review Comments:

Regarding data sets, each salvianolic acid is showing three types of events. How the signature signal of bioactive compounds is evaluated during database preparation.

Author Response:

We thank reviewer 2 for this constructive comment on signature signal evaluation during the database building. For the analytes reporting multiple event types, such as SalB and SalA, the data that corresponds to different types of events is not classified to specific types before being used for training. Since our sensing purpose is to judge whether an event belongs to a specific analyte instead of a specific type of the analyte, a pre-classification of data is not required.

We however find this inquiry raised by reviewer 2 extremely useful. To avoid causing a similar misunderstand to other readers, some relevant discussions were added in the revised manuscript to clarify this technical detail. It is also pasted below for the ease of your reference.

“Different event types generated by the same type of salvianolic acid were not differently labelled during the training process.”

Review Comments:

Why only two features other than ionic current blockage(I₀) are considered? Other than $\Delta I/I_0$ and S.D, what will be the effect of other mathematical operations (logI, I_{min}/I, I_{mx}/I etc.) on the ML classification results?

Author Response:

We thank reviewer 2 for raising this valuable discussion of the feature selection. First of all, our choice of event features is based on the most apparent differences between different types of events, as shown in the 2D scatter plot of $\Delta I/I_0$ and *S.D.* (**Figure 2i**). These two event parameters are also widely used in other nanopore studies by us (*Angew. Chem. Int. Ed.* 61, e202203769 (2022). <https://doi.org/10.1002/anie.202203769>; *Nat. Nanotech.* 17, 976–983 (2022). <https://doi.org/10.1038/s41565-022-01169-2>) and others (*ACS Nano* 16, 7258-7268 (2022). <https://doi.org/10.1021/acsnano.1c11455>; *Angew. Chem. Int. Ed.* 61, e202209970 (2022). <https://doi.org/10.1002/anie.202209970>). ΔI and $\Delta I/I_0$ are linearly correlated event parameters, the use of either one will produce the same result. However, the discrimination of SalB, LSA and RA can not be achieved only relying on ΔI , without the assistance of *S.D.* (**Figure RL 2.4**)

The reviewer is correct that other event features can also be further included in the machine learning model. We didn't do that in this manuscript because the existing model is already performing sufficiently well and a 99.0% accuracy is reported.

We also thank your suggestion of other event parameters, however, logI and I are strongly correlated and should not be simultaneously used in the same model. For I_{min}/I and I_{max}/I, these two parameters both may be significantly interfered by random and transient spiky noises on top of the blockage level. Thus, they are rarely used in nanopore analysis.

However, some parameters such as the skewness, the dwell time and the kurtosis can as well be introduced to the model in the future. Eventually, we again thank reviewer 2 for raising such useful discussions. To fully address this issue, some relevant discussions of the selection of event parameters have been added to the revised manuscript, as pasted below.

*“Though only two event parameters, including $\Delta I/I_0$ and *S.D.* were used in the machine learning model building, the model performance is satisfying, suggesting that the raw data is of a high quality and data separation. In the future, when more analytes were to be simultaneously analysed, more event parameters, such as skewness, kurtosis*

and dwell time may be further included.”

Review Comments:

What is role of concentration of bioactive molecules on the ML classification, especially for CA, PCA, and PA molecules.

Author Response:

We acknowledge reviewer 2 for raising this constructive discussion on the influence of the analyte concentration on ML. During nanopore sensing, a higher concentration of the analyte will produce a higher rate of event appearance, meaning that the events will be more efficiently acquired. An example is demonstrated with the concentration dependence measurement using CA (**Figure RL 2.3**). However, acknowledging the single molecule nature of this measurement, the event features are not affected by the analyte concentration at all. Thus, the choice of the analyte concentration won't influence the model building at all. However, a suitably high analyte concentration is helpful to more efficiently gain model events for data training.

We acknowledge reviewer 2 again for this inspiring inquiry. Some relevant discussions have been added to the revised manuscript to thoroughly clarify this point.

“The event features are also independent of the concentration of the analyte used to produce the event.”

Figure RL 2.3. The event features of CA at different concentrations. Taking CA as an example, the event features are independent of the analyte concentration. However, the event appearance rate is clearly increased when a higher analyte concentration is used.

Review Comments:

The author argued, “To avoid human judgemental bias,” the ML method is applied. How is ML classification assisting or accelerating bioactive molecule identifications? Since data cleaning, denoising, and feature extraction are complex processes, any slight aberration may mislead the prediction.

Author Response:

Thanks reviewer 2 for this comment on machine learning process. The machine learning is assisting or accelerating bioactive molecule identification in following ways:

We however find the introduction of the ML algorithm more advantageous than visual inspection in these aspects of view.

- 1. The automation.** The whole process of machine learning assisted event identification, including feature extraction, model training and model prediction et al, can be automatically performed with the custom Python-based algorithms. This is extremely useful when a large quantity of data was generated. However, it is extremely time-consuming and labor-intensive for humans to do that manually and visually.
- 2. The speed of the analysis.** Specifically, for herbal medicine analysis, the whole automated data analysis can be finished in just ~min using machine learning algorithms, which is not easily achieved solely by human judgment.
- 3. Further introduction of multiple event features.** In this work, we are satisfied with the performance achieved by simultaneously considering two event features, the relative blockage depth ($\Delta I/I_0$) and the standard deviation (*S. D.*). However, this is only for simultaneous discrimination of eight salvianolic acids. Soon, when more analytes were further introduced to this system, more events parameters including, skewness (skew), kurtosis (kurt), dwell time and many others may be further included to fit the sensing need. The establishment of the current ML algorithm is technically prepared for that.
- 4. A quantitative and objective judgment of the discrimination accuracy.** The machine learning algorithm provides a systematic, quantitative and objective measurement of the data discrimination performance. These are based on results of the cross-validation accuracy, confusion matrix and the learning curves. However, it is hard to draw quantitative conclusion only by visual judgement of human.
- 5. The preparation for noise background events.** For complex natural samples, such as real natural herbs (**Figure S24**), many interfering events can seriously affect visual judgment, which is hard for humans to judge. The introduction of machine learning algorithms can automatically remove interfering events to guarantee the

robustness of data analysis against noise background events.

Eventually, we thank reviewer 2 again for raising this insightful discussion of the advantages of using machine learning. We believe that the above discussions could address your inquiry. Some relevant discussions have also been added to the revised manuscript as well.

“In this work, the data acquired with complex samples leads to the difficulty of events identification by human eyes. Besides, when large amount of data is involved, event identification automation also becomes urgent. To also quantitatively assist event identification and sensing performance evaluation, a Python-based machine learning algorithm was developed.”

“Though only two event parameters, including $\Delta I/I_0$ and S.D. were used in the machine learning model building, the model performance is satisfying, suggesting that the raw data is of a high quality and data separation. In the future, when more analytes were to be simultaneously analyzed, more event parameters, such as skewness, kurtosis and dwell time may be further included.”

Review Comments:

What could be the reason behind the best performance of simpler KNN models than RF and XGBoost models with these data sets? What are the hyperparameters considered?

Author Response:

We acknowledge reviewer 2 for this comment. According to **Figure 3b**, all models are performing well and the difference between different machine learning models are almost negligible. I believe it is because the events generated by different analytes are already fully separated. Thus, the choice of model thus becomes indifferent. For this reason, all models were performing quite well. Since a satisfying accuracy is already achieved, we didn't attempt to further modify the hyperparameters. Only default hyperparameters were used.

It is also extremely hard to interpret exactly why a model is performing slightly better than the other. Since only a ~0.1% difference is reported for different models and all models are performing, we find it not that urgent to further investigate this issue. We also apology that we are not an expert of computer science and this is slightly out of our expertise. To fully address your inquiry, some relevant discussions were also added to the revised manuscript, as pasted below.

“All models report a high classification accuracy, suggesting that the data acquired with different salvianolic acids was easily discriminable. All above trainings were carried out with the default hyperparameter settings.”

Review Comments:

Can the optimized ML models are integrated with the experimental setup for automatic identification of molecules. How feasible and efficient it would be?

Author Response:

We thank reviewer 2 for raising this highly prospective suggestion. I think it is a good and rather feasible idea to integrate the ML model into the experimental setup in the near future. However, it is not the task, at least in this paper. A potential plan is to integrate the ML code into a highly portable nanopore sensing device, so that the whole device may be fully portable and used anywhere. This will be extremely useful for the analysis of natural herb samples. However, this demonstration can be shown in a future separate work instead. Still, it is a great suggestion, the relevant discussion is also added to the revised manuscript to fully address this issue as well.

“This sensing principle can also be further integrated into a portable device to assist natural product investigations in the field or in extreme situations when access to state-of-the-art instruments becomes impossible.”

Review Comments:

Can author comment on which feature is mainly driving the better classification results?

Author Response:

We acknowledge reviewer 2 for this constructive discussion. During machine learning, two features, $\Delta I/I_0$ and $S.D.$, were employed for the event classification. Besides, some salvianolic acids are indistinguishable when either $\Delta I/I_0$ or $S.D.$ was considered. **This is clearly shown with results of SalB, SalA and RA (Figure RL 2.4).**

Furthermore, the corresponding mutual information between the two features and the classes are calculated and described in **Figure RL 2.4** (now **Figure S18** in the revised manuscript). The mutual information measures the correlation between the features and event labels, where a higher value indicates a closer correlation. By calculating the mutual information between the features and target labels, we can determine which features are more important for the prediction. According to the calculation in **Figure RL 2.4**, the values of the two features are similar, **further illustrating the equal importance of $\Delta I/I_0$ and $S.D.$ for the identification results.** The relevant discussion is also added to the revised manuscript to fully address this issue as well.

“Considering the influence of different features on the prediction results, the importance of $\Delta I/I_0$ and $S.D.$ are evaluated for eight analyte discrimination. The

superimposed histograms of SalB, SalA and RA were shown in Figure S18 and demonstrated the fact that some salvianolic acids are indistinguishable when either $\Delta I/I_0$ or S.D. was employed. Furthermore, the corresponding mutual information between the two features and the classes were also calculated and described in Figure S18. The mutual information value measures the correlation between the features and event labels, where a higher value indicates a closer correlation and the more important this feature is. Obviously, the values of the two features are similar, further illustrating the equal importance of $\Delta I/I_0$ and S.D. for identification results.”

Figure RL 2.4. The comparison of parameter importance for eight salvianolic acid identification. (a) The superimposed histogram of $\Delta I/I_0$ of events acquired with SalB and SalA. (b) The superimposed histogram of S.D. of events acquired with SalB and RA. **Clearly, event overlaps are clearly seen when only a single event parameter is used.** (c) The mutual information values of $\Delta I/I_0$ and S.D..

Review Comments:

Minor Comments:

As the methods section is given in the main manuscript, the authors should mention it as methods only not as SI methods as given in line no.88.

Author Response:

We acknowledge reviewer 2 for this friendly reminder. We have performed all relevant corrections, strictly following your request.

Review Comments:

Figure citations are not in order. Should Fig 1b be cited before Fig 1a?

Author Response:

We highly appreciate reviewer 2 for pointing out this figure citation issue. We have performed all relevant revisions as requested in the revised manuscript.

Review Comments:

Throughout writing of the manuscript can be improved.

Author Response:

We thank reviewer 2 for this comment and we also apologize for the relevant writing issues of the manuscript. We have revised the manuscript accordingly. The manuscript has also been seen by professional manuscript editors as well.

REVIEWER COMMENTS

Reviewer #1 (Remarks to the Author):

The authors fully improved the manuscript and addressed my questions satisfactorily. I think that the manuscript is ready for publication in Nature Communications.

Reviewer #3 (Remarks to the Author):

In this study, PBA-modified MspA pores were employed to detect eight types of salvianolic acid molecules. During the translocation of these molecules, their 1,2-diol groups exhibit specific interactions with the functionalized PBA on the nanopore wall. Consequently, the translocation time is significantly extended, and these specific interactions manifest as fingerprint-like patterns in the ionic current, aiding in the detection and identification of different molecules.

The technique was applied to identify various salvianolic acids present in herbal medicines, including salvianolate injection and natural herbs. Utilizing straightforward machine learning algorithms enables automatic, rapid, and accurate identification in both qualitative and quantitative aspects. Overall, this work is comprehensive, yielding consistent and promising results. . This is a good attempt of the real-life application of biological nanopore sensing technology. However, there is room for further improvement in certain aspects of the manuscript:

1. The title appears overly broad, as the study specifically focuses on identifying salvianolic acids in three herbs. I recommend a more specific title to accurately reflect the scope of the work.
2. Following feature extraction from translocation events, the authors employed DBSCAN to clean the data by excluding points away from all clusters. This treatment requires further justification: a.) Clarify possible reasons for these excluded events—could they originate from sample impurities, non-specific interactions, or system noise? b.) Provide a detailed discussion on the ratios of excluded data points. Although the authors already gave the number of excluded data point in SI, I think it is better to calculate the ratios for different samples/conditions, and discusses them in the main text. For example, it seems that different salvianolic acids show different excluded ratios. What are the possible reasons? Does this related to the molecule structures and the PBA-molecule interactions? Furthermore, it is interesting to compare the excluded ratios of pure samples and real herbal medicine samples. It seems that there are more data points are excluded in the herbal samples. So, what are the possible reasons?
3. In the concluding section of this manuscript, it would be intriguing to briefly explore the potential and challenges of the technology. Consider discussing the feasibility of modifying the pore with other chemical groups, introducing diverse molecular interactions, and extending the application to detect/identify other chemical molecules. A general roadmap and corresponding challenges for such advancements could be outlined.
4. Regarding the experiments, as the concentration of molecules increases, the on-time (τ_{on}) becomes shorter and shorter, which means that the events are closer to the adjacent ones. If the concentration increases further, the events may be not independent from others, indicating the current

translocation event may influence the next coming events, or even multiple molecules can be captured and stay inside the nanopore at the same time. Did the authors find similar influences from the extra high concentration in the experiments? What concentration range can this technology effectively cover?

5. The manuscript emphasizes the quantitative nature of nanopore analysis, and the authors have compared extracted concentrations of related molecules. However, in the case of natural herb samples, the authors only claimed that the content distributions are consistent with the results from previous investigations using HPLC from reference 73. In this reference, only the concentrations of various contents can be found in tables. I still have no idea about how good the results from nanopore and HPLC match with each other. To provide a clearer understanding of the correlation between nanopore and HPLC results, I recommend a more detailed and quantitative comparison. This will give us a general feeling about accuracy of the nanopore results by benchmarking against the well-established HPLC method.

In general, the authors' responses to Reviewer 2 are scientifically sound and relevant. They have provided key information to address the reviewer's questions, and the modifications in the manuscript and supporting information are both relevant and reasonable. However, some responses could benefit from being more concise and directly addressing the points raised. If you would like a score, I will give 80/100 for these responses. To easy locate, I have numbered the comment-response pairs sequentially in the review report.

The reviewer's comments can be divided into the following several categories:

- Summary: #1
- Writing: #2
- Sensing mechanisms: #3, #4, #5, #6, #7, #10
- ML algorithms: #8, #9, #11, #12, #13, #14, #15
- Outlook: #16, #17
- Miner points: #18, #19, #20

1. The reviewer asked several time for the reasons of the high sensitivity of sensing, for example #6 and #10. The key factors contributing to this sensitivity are as follows: 1. Although the size difference among molecules (CA, PAC, and PA) is small, variations in chemical environment result in a.) different kon-koff rates in the diol-PBA reaction (related to S.D.), b.) distinct additional interactions with amino acid side groups on the pore wall (related to SD and $\Delta I/I_0$), and c.) diverse ion distributions near the pore restriction affecting ionic current (related to $\Delta I/I_0$). 2. The nanopore restriction size is in the nano-meter range, comparable to target molecule size, with signals primarily determined by interactions/local physio-chemical environment, leading to extreme sensitivity. The authors delivered this information but not very straightforward and concise.

2. In response to comment #7, the reviewer inquired about the nanopore's capability to identify trans-diols. The authors presented experimental results and described the sensing performance on trans-diols but did not explicitly explain the significant difference between the sensing performance of cis- and trans-diols. The possible reasons for this substantial difference are straightforward: cis and trans isomers

have different spatial conformations of the diol group. One matches the spatial conformation of PBA, while the other does not.

Response Letter

Reviewer #1

Remarks to the Author:

The authors fully improved the manuscript and addressed my questions satisfactorily. I think that the manuscript is ready for publication in Nature Communications.

Author Response:

We are extremely grateful for reviewer 1 for considering that we have fully and satisfactorily addressed all your review reports. Your recommendation to publish this work is also highly appreciated.

Reviewer #3

Remarks to the Author:

In this study, PBA-modified MspA pores were employed to detect eight types of salvianolic acid molecules. During the translocation of these molecules, their 1,2-diol groups exhibit specific interactions with the functionalized PBA on the nanopore wall. Consequently, the translocation time is significantly extended, and these specific interactions manifest as fingerprint-like patterns in the ionic current, aiding in the detection and identification of different molecules.

The technique was applied to identify various salvianolic acids present in herbal medicines, including salvianolate injection and natural herbs. Utilizing straightforward machine learning algorithms enables automatic, rapid, and accurate identification in both qualitative and quantitative aspects. Overall, this work is comprehensive, yielding consistent and promising results. This is a good attempt of the real-life application of biological nanopore sensing technology. However, there is room for further improvement in certain aspects of the manuscript:

Author response:

We would like to thank reviewer 3 for an accurate description of the main topic of this manuscript and your recognition of the nanopores applications in real-life samples as a “good attempt” is highly appreciated. Your supportive comments on the high resolution of the MspA-90PBA to specifically discriminate between multiple salvianolic acids and the identification capacity of machine learning to implement the qualitative and quantitative analysis of herbal medicines are truly appreciated. We are also grateful for reviewer 3 for considering this work “comprehensive, consistent and promising”. We also sincerely acknowledge reviewer 3 for your other valuable comments on this manuscript. Detailed responses to your comments are listed below in a point by point style, for the ease of your reference.

Remarks to the Author:

1. The title appears overly broad, as the study specifically focuses on identifying salvianolic acids in three herbs. I recommend a more specific title to accurately reflect the scope of the work.

Author Response:

We acknowledge reviewer 3 for raising this insightful discussion on the title. In the revised manuscript, the title has been changed to “**Nanopore analysis of salvianolic acids in herbal medicines**”. This should well describe what we have done in this manuscript and your inquiry should have been fully addressed.

Remarks to the Author:

2. Following feature extraction from translocation events, the authors employed DBSCAN to clean

the data by excluding points away from all clusters. This treatment requires further justification: a.) Clarify possible reasons for these excluded events—could they originate from sample impurities, non-specific interactions, or system noise? b.) Provide a detailed discussion on the ratios of excluded data points. Although the authors already gave the number of excluded data point in SI, I think it is better to calculate the ratios for different samples/conditions, and discusses them in the main text. For example, it seems that different salvianolic acids show different excluded ratios. What are the possible reasons? Does this related to the molecule structures and the PBA-molecule interactions? Furthermore, it is interesting to compare the excluded ratios of pure samples and real herbal medicine samples. It seems that there are more data points are excluded in the herbal samples. So, what are the possible reasons?

Author Response:

We acknowledge reviewer 3 for the valuable and insightful comments on the discussion of the ratios of the removed events. Technically, the interference events removed by DBSCAN are non-clustered events, meaning that these events don't have consistent event feature appearances when probed by nanopores. They may originate from impurities in the sample, non-specific interaction of the analyte with the pore or spontaneous gating of the nanopore itself. Actually, a variety of non-clustered events as described above even appear before the addition of the analyte, suggesting that they are not relevant to the analyte of interests at all. For example, the spontaneous pore gating events and the inherent short residing noises introduced by the pore itself. For analytes that report non-specific interaction with the PBA adapter, their nanopore events are generally short-residing and non-clustered. That is why they could be removed by DBSCAN.

Moreover, salvianolic acids can also be chemically unstable (*Journal of Pharmaceutical and Biomedical Analysis* 43, 435-439 (2007); *International Research Journal of Pure & Applied Chemistry* 7, 99-109 (2015)). They may spontaneously undergo chemical degradation during the measurements. According to reports in the literature, SAA and SalA are not stable in aqueous solution (*Spectrochimica Acta Part A: Molecular and Biomolecular Spectroscopy* 78, 1535-1539 (2011); *Pharmacogn Mag* 9, 338-343 (2013)). The degraded compounds may as well produce short-residing and randomly scattering dots in the data as well. Though it is not the focus of this study, we have also included the discussion of this technical possibility in the revised manuscript as well.

Besides DBSCAN, the One-Class SVM algorithm, which detects outlier events, is also used for data cleaning for herbal medicines. Here, the outlier events are events that don't resemble to any previously recorded events acquired with standard salvianolic acids compounds. These events may originate from unknown impurities in the sample which can also react with the PBA adapter to produce events. Though these outlier events may appear as clear clusters. These events, which are not of interests by us, are removed by One-Class SVM.

All above mentioned events were removed by pre-established computer algorithms. "Real-life" samples inevitably contain impurities which may generate measurement noises of this kind. Here, the beauty of a high resolution nanopore sensor and the AI algorithm is that they efficiently minimize the interferences caused by these impurities.

Also following that requested by reviewer 3, we have listed the ratios of the removed events in **Table RL1.1 (Table S2** in the revised manuscript). The reviewer is correct that there are more events removed from natural herb samples. **It is expected because the natural herb extracts contain a large variety of cis-diol containing compounds which may also generate events.** According to previous literature studies of these herbs, the interferences could be from saccharides (*TrAC Trends in Analytical Chemistry* 52, 155-169 (2013)), anthocyanin (*Journal of Medicinal plants and By-product* 7, 163-171 (2019)) and many others. These interfering compounds are not of our interests in this study so these events were computationally removed by computer algorithms. Acknowledging the high resolution of nanopore, these events are significantly different in the event features than those produced by standard salvianolic acids compounds.

Eventually, to fully address your inquiry, the addition of the corresponding table and relevant discussions have been added to the revised manuscript. We would like to acknowledge reviewer 3 again for raising these discussions, which have further improved this manuscript.

For the ease of your references, the added table and the discussions are also pasted below as well.

Table RL1.1. The interference events ratios of standard analytes and herbal medicines. Events acquired with pure standard analytes and herbal medicines were performed with corresponding data pre-treatments for interference events removal. The ratio of removed events was summarized below.

Salvianolic Acids and Natural Samples	Ratio of interference events (%)
CA	5.7 ± 1.3
PCA	5 ± 2
PA	2.9 ± 0.9
SAA	18.3 ± 1.9
RA	10.8 ± 1.1
LSA	12.5 ± 0.4
SalA	30 ± 2
SalB	7.1 ± 1.1
salvianolate injection	21.4 ± 0.6
Salvia miltiorrhiza	31 ± 4
Rosemary	71 ± 4
P. vulgaris	78 ± 6

“The ratios of interference events removed by DBSCAN are summarized in Table S2. The interference events can be from impurities in the analytes derived from plant extraction or chemical degradation of salvianolic acids. Spontaneous pore gating also contributes to the generation of interference events as well.”

“The interference events, which don’t resemble to any events previously reported by the eight standard salvianolic acids, were removed by One-Class SVM (Figures S26-S27). Compared with

standard analytes (Table S2), nanopore measurements performed with natural herb extracts report more interference events. It is expected because a variety of cis-diols in natural herbs such as saccharides and anthocyanin may also bind to the PBA adapter to generate nanopore events.”

Remarks to the Author:

3. In the concluding section of this manuscript, it would be intriguing to briefly explore the potential and challenges of the technology. Consider discussing the feasibility of modifying the pore with other chemical groups, introducing diverse molecular interactions, and extending the application to detect/identify other chemical molecules. A general roadmap and corresponding challenges for such advancements could be outlined.

Author Response:

We acknowledge reviewer 3 for this valuable and prospective suggestion. The reviewer is correct that though demonstrated with PBA adapter using a hetero-octameric MspA nanopore, this system is open for a wide variety of other sensing moieties in the future. With the increase of the system complexity and the resolution of the pore, the use of machine learning and the corresponding data base are becoming indispensable for automated data analysis as well. The above-mentioned system may also be further integrated into a highly compact device to achieve a much-lowered detection limit or a highly portable instrument size as well.

Eventually, we would like to thank reviewer 3 for raising the discussion of the technical prospects of this system. All relevant discussions have been added to the revised manuscript. For the ease of your reference, they are also pasted below:

“To further expand its sensing capacity, this hetero-octameric MspA may be installed with other reactive adapters, including those based on coordination chemistry, disulfide chemistry or click chemistry, so that more diverse types of analytes may be sensed. With the increased complexity of the generated event features, the use of machine learning by simultaneous consideration of more event features or deep learning becomes indispensable. The whole setup may as well be further integrated into a miniaturized chip and used with a highly portable device, for applications in the field or in extreme situations when access to state-of-the-art instruments becomes impossible.”

Remarks to the Author:

4. Regarding the experiments, as the concentration of molecules increases, the on-time (τ_{on}) becomes shorter and shorter, which means that the events are closer to the adjacent ones. If the concentration increases further, the events may be not independent from others, indicating the current translocation event may influence the next coming events, or even multiple molecules can be captured and stay inside the nanopore at the same time. Did the authors find similar influences from the extra high concentration in the experiments? What concentration range can this technology effectively cover?

Author Response:

We acknowledge reviewer 3 for this constructive and insightful comment on high concentration measurements. The reviewer is correct that when the concentration of the analyte is getting higher, the inter-event interval between adjacent events are becoming shorter. **However, according to this specific pore design, only a sole reactive site (PBA) was introduced to the pore constriction and thus only a single analyte may be simultaneously bound to the PBA site.** This is also why we need to build a hetero-octameric porin to perform this measurement.

Figure RL 1.1. The mechanism of CA sensing. (a) The structure of MspA-90PBA. A single phenylboronic acid (PBA) adapter was modified at its constriction. (b) The mechanism of salicylic acids sensing using MspA-90PBA. **MspA-90PBA has only one *cis*-diol group binding site at its constriction, indicating that only one molecule can be captured simultaneously.**

Experimentally, when the only PBA site in the pore is occupied with a *cis*-diol molecule, further binding of a secondary *cis*-diol is technically prohibited. **Thus, simultaneous binding of two *cis*-diols to the same pore is not technically possible (Figure RL1.1), even at a high analyte concentration.** In another word, this measurement is “single molecule” not only because the pore is sufficiently narrow but also because there is only a sole reactive site placed at the pore constriction. **Thus, the event features are independent of the analyte concentration.** One clear example is shown in **Figure RL1.2**. The event features of CA are almost identical, even if the measurements were respectively acquired at low and high CA concentrations.

Figure RL 1.2. The CA sensing at different concentrations. Representative traces of CA at low (25 μM) and high (1000 μM) concentrations were presented in the figure. The events in the red dotted boxes were individually enlarged and exhibited, indicating that the event features are independent of the analyte concentration.

The reviewer is correct that the concentration dependence would be different when the measurement was carried out at an extremely high analyte concentration (**Figure RL1.3**). Taking SalB as an example, at a lower analyte concentration, the concentration dependence of the derived $\overline{1/\tau_{on}}$ value is generally linear. However, at a high concentration, the concentration dependence is getting saturated and the derived quantitative value is getting inaccurate. This indicates that the concentration range that this technology can effectively cover is the concentration range that the concentration dependence is still linear (**Figure RL1.4**). **To fully address this issue, an SI table describing the effective concentration dependence range is provided.** Thus, for samples containing analyte out of this concentration range, analyte enrichment or dilution would be necessary solve this issue.

Figure RL 1.3. The value change of $1/\tau_{on}$ at extremely high analyte concentrations. (a) The correlation between the $1/\tau_{on}$ and concentrations of SalB. The $1/\tau_{on}$ values did not change much when the concentration exceeds 225 μM. (c) Representative trace of SalB at a final concentration of 25 μM and 450 μM. The trace was taken from a 10 s continually recorded.

Table RL1.2. The effective concentration dependence ranges of eight salvianolic acids. The effective concentration ranges for eight salvianolic acids were determined by the linear concentration dependence of the $\overline{1/\tau_{on}}$ value as described in **Figure RL 1.4**. The corresponding ranges were also summarized below.

Salvianolic Acids and Natural Samples	Effective concentration range (μM)
CA	25 ~ 500
PCA	50 ~ 2000
PA	5 ~ 400
SAA	50 ~ 1500
RA	2 ~ 30
LSA	25 ~ 300
SalA	5 ~ 40
SalB	25 ~ 125

Figure RL 1.4. The effective concentration dependence of eight salvianolic acids sensing. The plot of $1/\tau_{on}$ versus the each salvianolic acid concentration. The $1/\tau_{on}$ is linearly correlated with each analyte over a wide concentration range, determining the effective concentration range for quantitative analysis. The error bars show standard deviations derived from results of three independent measurements (N=3).

We again sincerely thank reviewer 3 for this valuable comment. To fully address this issue, some relevant discussion of linear response was added to the revised manuscript.

“According to the nanopore design, measurements performed at extremely high concentrations of analyte would result in the saturation of the PBA adapter and report inaccurate quantification. Thus, the effective concentration range for this measurement is defined to be the range of analyte concentration within which the $1/\tau_{on}$ is linearly correlated with the input analyte concentration (Figure S31, Table S8). For measurements beyond the effective concentration range, sample enrichment or dilution will become necessary.”

Remarks to the Author:

5. The manuscript emphasizes the quantitative nature of nanopore analysis, and the authors have compared extracted concentrations of related molecules. However, in the case of natural herb samples, the authors only claimed that the content distributions are consistent with the results from previous investigations using HPLC from reference 73. In this reference, only the concentrations of various contents can be found in tables. I still have no idea about how good the results from nanopore and HPLC match with each other. To provide a clearer understanding of the correlation between nanopore and HPLC results, I recommend a more detailed and quantitative comparison. This will give us a general feeling about accuracy of the nanopore results by benchmarking against the well-established HPLC method.

Author Response:

Thanks reviewer 3 for raising the suggestion for a detailed quantitative compare of natural herb analysis performed using different platforms. Following your requests, a table which clearly describes natural herb analysis however performed by other platforms were included in **Table RL1.3**. However, since we are not experts of HPLC or HPLC-MS, we can only compare results reported in literatures with our results. Please also note that the source of the herb and the extraction methods in these literatures are not completely identical to ours. However, according to the table, the reported quantitation results are generally close to that reported by that measured by other platforms. Since our extraction method are gentler and doesn't involve any heating or organic solvent extraction, it is expected that our results would be slightly lower than that reported in previous literatures.

Natural herbs	Content (mg/g)	Materials source	Methods	References
Salvia miltiorrhiza (SalB)	12 ± 4 (our work)	Juxian, Shandong Province of China	water for extraction, soaking for 16 h	
	4.77–40 (HPLC)	Linqu County, Shandong Province of China	70% aqueous methanol for extraction, ultrasound for 20 min	Molecules 17, 2388-2407 (2012)
	17.92–34.8 (HPLC)	Laiwu City, Shandong Province of China	water for extraction, reflux for 4 h	Herald of Medicine 38, 1624-1629 (2019)
	~20 (HPLC)	Shangluo, Shaanxi Province of China	water for extraction, ultrasound for 30 min, ethyl acetate reextraction for 5 times	Ultrasonics Sonochemistry 17, 61-65 (2010)
Rosemary (RA)	1.26 ± 0.08 (our work)	Bozhou City, Anhui Province of China	water for extraction, soaking for 16 h	
	0.014–6.5 (HPLC)	Peñafiel (Valladolid, Spain)	pretreatment with deoiled, deoiled + milled or fresh leaves, water for extraction at 40 °C for 4 h	Journal of Food Engineering 109, 98-103 (2012)
	0.9–41.8 (HPLC-MS)	Murcia in Spain	Methanol extraction	Journal of Chromatography A 1120, 221-229, (2006)
P. vulgaris (RA)	0.53 ± 0.12 (our work)	Bozhou City, Anhui Province of China	water for extraction, soaking for 16 h	
	1.19–9.46 (HPLC)	Turkey	water for extraction with HCl and ascorbic acid addition for 8h stirring	Journal of Pharmaceutical and Biomedical Analysis 55, 1227-1230 (2011)
	2–7 (HPLC)	China	methanol for extraction, reflux for 1 h	Chinese Traditional and Herbal Drugs 51, 2842-2850 (2020)

Table RL1.3. Comparison between our results and relevant researches on salvianolic acids content in natural herbs. SalB content in *Salvia miltiorrhiza* and RA content in *Rosemary* and *P. vulgaris* reported in both our work and relevant literatures were demonstrated in the list below, respectively. Moreover, the corresponding materials source, extraction or pretreatment methods were also summarized.

To also fully address reviewer 3's inquiry, the corresponding tables (**Table S5** and **Table S6**) and relevant discussions have been added to the revised manuscript. We again thank reviewer 3 for this valuable comment. As suggested, the added discussion should be helpful for the readers as well. For the ease of reference, the added discussion is also pasted here:

*“The $1/\tau_{on}$ values of SalB events were subsequently calculated as described in Methods, according to which, the amount of SalB extracted from *Salvia miltiorrhiza* was derived based on the calibration curve (Table S5). Finally, the SalB in *Salvia miltiorrhiza* was calculated to be $\sim 12 \pm 4$ mg/g, which is highly consistent with those reported in previous literatures (Table S6).”*

“The calibration curve of RA, demonstrating a coefficient factor ($R^2 = 0.995$) is shown in Figure

S30 and Table S6. The corresponding $1/\tau_{on}$ values of RA events acquired with Rosemary and P. vulgaris were measured (Methods) and summarized in Table S5. Afterwards, based on the calibration curve, the RA concentration in Rosemary and P. vulgaris were derived to be $\sim 1.26 \pm 0.08$ mg/g and $\sim 0.53 \pm 0.12$ mg/g (Table S5), which are generally consistent with those previously investigated by HPLC (Table S6). The differences of RA concentration in our work and that reported in literatures may be due to the different material resource, sample pre-treatments and extraction processes”

Remarks to the Author:

In general, the authors' responses to Reviewer 2 are scientifically sound and relevant. They have provided key information to address the reviewer's questions, and the modifications in the manuscript and supporting information are both relevant and reasonable. However, some responses could benefit from being more concise and directly addressing the points raised. If you would like a score, I will give 80/100 for these responses. To easy locate, I have numbered the comment-response pairs sequentially in the review report.

The reviewer's comments can be divided into the following several categories:

- Summary: #1
- Writing: #2
- Sensing mechanisms: #3, #4, #5, #6, #7, #10
- ML algorithms: #8, #9, #11, #12, #13, #14, #15
- Outlook: #16, #17
- Miner points: #18, #19, #20

1. The reviewer asked several time for the reasons of the high sensitivity of sensing, for example #6 and #10. The key factors contributing to this sensitivity are as follows: 1. Although the size difference among molecules (CA, PAC, and PA) is small, variations in chemical environment result in a.) different kon-koff rates in the diol-PBA reaction (related to S.D.), b.) distinct additional interactions with amino acid side groups on the pore wall (related to SD and $\Delta I/I_0$), and c.) diverse ion distributions near the pore restriction affecting ionic current (related to $\Delta I/I_0$). 2. The nanopore restriction size is in the nano-meter range, comparable to target molecule size, with signals primarily determined by interactions/local physio-chemical environment, leading to extreme sensitivity. The authors delivered this information but not very straightforward and concise.

2. In response to comment #7, the reviewer inquired about the nanopore's capability to identify trans-diols. The authors presented experimental results and described the sensing performance on trans-diols but did not explicitly explain the significant difference between the sensing performance of cis- and trans-diols. The possible reasons for this substantial difference are straightforward: cis and trans isomers have different spatial conformations of the diol group. One matches the spatial conformation of PBA, while the other does not.

Author Response:

We sincerely acknowledge reviewer 3 for confirming the quality and the accuracy of our response to reviewer 2 by stating that our response “scientifically sound and relevant” and the corresponding revisions are “both relevant and reasonable”. We also acknowledge your professional and clear

summary of our response as well.

We also acknowledge your suggestion to more clearly explain why this nanopore is having a high resolution against molecular targets with variable spatial sizes. Acknowledging your suggestion, we have included the added discussion in the conclusion session of the revised manuscript. It is also pasted below for the ease of your reference:

“This high-resolution results from the sufficiently narrow pore constriction, which is comparable to target molecule size. The event features are primarily determined by interactions/local physio-chemical environment, leading to the extreme high resolution of the pore.”

We also particularly acknowledge your comment on the capacity of this nanopore to detect *trans*-diols. In the revised manuscript, we clearly admit that this technique is not suitable for *trans*-diols. We are also pasting the revised sentence below for the ease of your reference:

“One drawback of this technique is that the PBA adapter is in principle not suitable for trans-diols due to the unmatched spatial conformation”

At this moment, we completely agree with all above listed technical summary of our response to reviewer 2 and our revision should be now more improved.

Eventually, we would like to thank reviewer 3 again for your time performing this review task and your precious revision suggestions which have further improved the overall quality of the manuscript as well. Your prompt feedback during the last round of revision, especially during this holiday season is highly appreciated. Merry Xmas and happy new year!!

REVIEWERS' COMMENTS

Reviewer #3 (Remarks to the Author):

The authors have fully addressed my questions and the responses are logical, scientific, and accurate. The manuscript has been adequately improved accordingly. Thus, I think the manuscript is ready for publication now.